# Risk-Sensitive Control as Inference with Rényi Divergence

**Kaito Ito**
The University of Tokyo
kaito@g.ecc.u-tokyo.ac.jp

**Kenji Kashima**
Kyoto University
kk@i.kyoto-u.ac.jp

## Abstract

This paper introduces the risk-sensitive control as inference (RCaI) that extends CaI by using Rényi divergence variational inference. RCaI is shown to be equivalent to log-probability regularized risk-sensitive control, which is an extension of the maximum entropy (MaxEnt) control. We also prove that the risk-sensitive optimal policy can be obtained by solving a soft Bellman equation, which reveals several equivalences between RCaI, MaxEnt control, the optimal posterior for CaI, and linearly-solvable control. Moreover, based on RCaI, we derive the risk-sensitive reinforcement learning (RL) methods: the policy gradient and the soft actor-critic. As the risk-sensitivity parameter vanishes, we recover the risk-neutral CaI and RL, which means that RCaI is a unifying framework. Furthermore, we give another risk-sensitive generalization of the MaxEnt control using Rényi entropy regularization. We show that in both of our extensions, the optimal policies have the same structure even though the derivations are very different.

## 1 Introduction

Optimal control theory is a powerful framework for sequential decision making [1]. In optimal control problems, one seeks to find a control policy that minimizes a given cost functional and typically assumes the full knowledge of the system's dynamics. Optimal control with unknown or partially known dynamics is called reinforcement learning (RL) [2], which has been successfully applied to highly complex and uncertain systems, e.g., robotics [3], self-driving vehicles [4]. However, solving optimal control and RL problems is still challenging, especially for continuous spaces.

Control as Inference (CaI), which connects optimal control and Bayesian inference, is a promising paradigm for overcoming the challenges of RL [5]. In CaI, the optimality of a state and control trajectory is defined by introducing optimality variables rather than explicit costs. Consequently, an optimal control problem can be formulated as a probabilistic inference problem. In particular, maximum entropy (MaxEnt) control [6, 7] is equivalent to a variational inference problem using the Kullback–Leibler (KL) divergence. MaxEnt control has entropy regularization of a control policy, and as a result, the optimal policy is stochastic. Several works have revealed the advantages of the regularization such as robustness against disturbances [8], natural exploration induced by the stochasticity [7, 9], fast convergence of the MaxEnt policy gradient method [10].

On the other hand, the KL divergence is not the only option available for variational inference. In [11], the variational inference was extended to Rényi $\alpha$-divergence [12], which is a rich family of divergences including the KL divergence. Similar to the traditional variational inference, this extension optimizes a lower bound of the evidence, which is called the variational Rényi bound. The parameter $\alpha$ of Rényi divergence controls the balance between mass-covering and zero-forcing effects for approximate inference [13]. However, if we use Rényi divergence for CaI, it remains unclear how $\alpha$ affects the optimal policy, and a natural question arises: what objective does CaI using Rényi divergence optimize?

38th Conference on Neural Information Processing Systems (NeurIPS 2024).

**Contributions** The contributions of this work are as follows:

1. We reveal that CaI with Rényi divergence solves a log-probability (LP) regularized risk-sensitive control problem with exponential utility [14] (Theorem 2). The order parameter $\alpha$ of Rényi divergence plays a role of the risk-sensitivity parameter, which determines whether the resulting policy is risk-averse or risk-seeking. Based on the result, we refer to CaI using Rényi divergence as *risk-sensitive* CaI (RCaI). Since Rényi divergence includes the KL divergence, RCaI is a unifying framework of CaI. Additionally, we show that the risk-sensitive optimal policy takes the form of the Gibbs distribution whose energy is given by the Q-function, which can be obtained by solving a soft Bellman equation (Theorem 3). Furthermore, this reveals several equivalence results between RCaI, MaxEnt control, the optimal posterior for CaI, and linearly-solvable control [15, 16].

2. Based on RCaI, we derive risk-sensitive RL methods. First, we provide a policy gradient method [17–19] for the regularized risk-sensitive RL (Proposition 7). Next, we derive the risk-sensitive counterpart of the soft actor-critic algorithm [7] through the maximization of the variational Rényi bound (Subsection 4.2). As the risk-sensitivity parameter vanishes, the proposed methods converge to REINFORCE [19] with entropy regularization and risk-neutral soft actor-critic [7], respectively. One of their advantages over other risk-sensitive approaches, including distributional RL [20, 21], is that they require only minor modifications to the standard REINFORCE and soft actor-critic. The behavior of the risk-sensitive soft actor-critic is examined via an experiment.

3. Although the risk-sensitive control induced by RCaI has LP regularization of the policy, it is not entropy, unlike the MaxEnt control with the Shannon entropy regularization. To bridge this gap, we provide another risk-sensitive generalization of the MaxEnt control using Rényi entropy regularization. We prove that the resulting optimal policy and the Bellman equation have the same structure as the LP regularized risk-sensitive control (Theorem 6). The derivation differs significantly from that for the LP regularization, and for the analysis, we establish the duality between exponential integrals and Rényi entropy (Lemma 5).

The established relations between several control problems in this paper are summarized in Fig. 1.

**Related work** The duality between control and inference has been extensively studied [15, 22–26]. Inspired by CaI, [27, 28] reformulated model predictive control (MPC) as a variational inference problem. In [29], variational inference MPC using Tsallis divergence, which is equivalent to Rényi divergence, was proposed. The difference between our results and theirs is that variational inference MPC infers *feedforward* optimal control while RCaI infers feedback optimal control. Consequently, the equivalence of risk-sensitive control

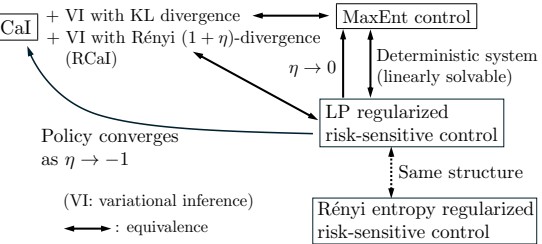

Figure 1: Relations of control problems.

and Tsallis variational inference MPC is not derived, unlike RCaI. The work [30] proposed an EM-style algorithm for RL based on CaI, where the resulting policy is risk-seeking. However, risk-averse policies cannot be derived from CaI by this approach. Our framework provides the equivalence between CaI and risk-sensitive control both for risk-seeking and risk-averse cases.

Risk-averse policies are known to yield robust control [31, 32], and risk-seeking policies are useful for balancing exploration and exploitation for RL [33]. Because of these merits, many efforts have been devoted to risk-sensitive RL [19, 34–36]. In [37], risk-sensitive RL with Shannon entropy regularization was investigated. However, their theoretical results are valid only for almost risk-neutral cases. Our results imply that LP and Rényi entropy regularization are suitable for the risk-sensitive RL.

In [16], risk-sensitive control whose control cost is defined by Rényi divergence was investigated, and it was shown that the associated Bellman equation can be linearized. However, it is assumed that the transition distribution can be controlled as desired, which is not satisfied in general as pointed out in [38]. On the other hand, our result shows that when the dynamics is deterministic, LP

and Rényi entropy regularized risk-sensitive control problems are linearly solvable without the full controllability assumption of the transition distribution.

**Notation**  For simplicity, by abuse of notation, we write the density (or probability mass) functions of random variables $x, y$ as $p(x), p(y)$, and the expectation with respect to $p(x)$ is denoted by $\mathbb{E}_{p(x)}$. For a set $S$, the set of all densities on $S$ is denoted by $\mathcal{P}(S)$. Rényi entropy and divergence with parameter $\alpha > 0$, $\alpha \neq 1$ are defined as $\mathcal{H}_\alpha(p) := \frac{1}{\alpha(1-\alpha)} \log\big[\int_{\{u:p(u)>0\}} p(u)^\alpha \mathrm{d}u\big]$, $D_\alpha(p_1\|p_2) := \frac{1}{\alpha-1} \log\big[\int_{\{u:p_1(u)p_2(u)>0\}} p_1(u)^\alpha p_2(u)^{1-\alpha} \mathrm{d}u\big]$. For the factor $\frac{1}{\alpha(1-\alpha)}$ of $\mathcal{H}_\alpha$, we follow [39, 40] because this choice is convenient for the analysis in Subsection 3.2 rather than another common choice $1/(1-\alpha)$. We formally extend the definition of $\mathcal{H}_\alpha$ to $\alpha < 0$. Denote the Shannon entropy and KL divergence by $\mathcal{H}_1(p)$, $D_1(p_1\|p_2)$, respectively because $\lim_{\alpha\to 1} \mathcal{H}_\alpha(p) = \mathcal{H}_1(p)$, $\lim_{\alpha\to 1} D_\alpha(p_1\|p_2) = D_1(p_1\|p_2)$. For further properties of the Rényi entropy and divergence, see e.g., [41]. The set of integers $\{k, k+1, \ldots, s\}$, $k < s$ is denoted by $[\![k, s]\!]$. A sequence $\{x_k, x_{k+1}, \ldots, x_s\}$ is denoted by $x_{k:s}$. The set of non-negative real numbers is denoted by $\mathbb{R}_{\geq 0}$.

## 2   Brief introduction to control as inference

First, we briefly introduce the framework of CaI. For the detailed derivation, see Appendix A and [5]. Throughout the paper, $x_t$ and $u_t$ denote $\mathbb{X}$-valued state and $\mathbb{U}$-valued control variables at time $t$, respectively, where $\mathbb{X} \subseteq \mathbb{R}^{n_x}$, $\mathbb{U} \subseteq \mathbb{R}^{n_u}$, and $\mu_L(\mathbb{U}) > 0$. Here, $\mu_L$ denotes the Lebesgue measure on $\mathbb{R}^{n_u}$. The initial distribution is $p(x_0)$, and the transition density is denoted by $p(x_{t+1}|x_t, u_t)$, which depends only on the current state and control input. Let $T > 0$ be a finite time horizon. CaI connects control and probabilistic inference problems by introducing *optimality variables* $\mathcal{O}_t \in \{0, 1\}$ as in Fig. 2. For

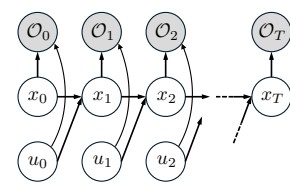

Figure 2: Graphical model for CaI.

$c_t : \mathbb{X} \times \mathbb{U} \to \mathbb{R}_{\geq 0}, c_T : \mathbb{X} \to \mathbb{R}_{\geq 0}$, which will serve as cost functions, the distribution of $\mathcal{O}_t$ is given by $p(\mathcal{O}_t = 1|x_t, u_t) = \exp(-c_t(x_t, u_t))$, $t \in [\![0, T-1]\!]$ and $p(\mathcal{O}_T = 1|x_T) = \exp(-c_T(x_T))$. If $\mathcal{O}_t = 1$, then $(x_t, u_t)$ at time $t$ is said to be "optimal." The control posterior $p(u_t|x_t, \mathcal{O}_{t:T} = 1)$ is called the optimal policy. Let the prior of $u_t$ be uniform: $p(u_t) = 1/\mu_L(\mathbb{U}), \forall u_t \in \mathbb{U}$. Although this choice is common for CaI, the arguments in this paper may be extended to non-uniform priors. Then, for the graphical model in Fig. 2, the distribution of the optimal state and control input trajectory $\tau := (x_{0:T}, u_{0:T-1})$ satisfies

$$p(\tau|\mathcal{O}_{0:T} = 1) \propto \left[p(x_0) \prod_{t=0}^{T-1} p(x_{t+1}|x_t, u_t)\right] \left[p(\mathcal{O}_T = 1|x_T) \prod_{t=0}^{T-1} p(\mathcal{O}_t = 1|x_t, u_t)\right]$$

$$= \left[p(x_0) \prod_{t=0}^{T-1} p(x_{t+1}|x_t, u_t)\right] \exp\left(-c_T(x_T) - \sum_{t=0}^{T-1} c_t(x_t, u_t)\right). \qquad (1)$$

For notational simplicity, we will drop $= 1$ for $\mathcal{O}_t$ in the remainder of this paper.

The optimal policy $p(u_t|x_t, \mathcal{O}_{t:T})$ can be computed in a recursive manner. To this end, define

$$\mathsf{Q}_t(x_t, u_t) := -\log \frac{p(\mathcal{O}_{t:T}|x_t, u_t)}{\mu_L(\mathbb{U})}, \quad \mathsf{V}_t(x_t) := -\log p(\mathcal{O}_{t:T}|x_t), \qquad (2)$$

which play a role of value functions. Then, the following result holds.

**Proposition 1.** *Assume that $\mu_L(\mathbb{U}) < \infty$ and let $\mathsf{c}_t(x_t, u_t) := c_t(x_t, u_t) + \log \mu_L(\mathbb{U})$. Assume further the existence of density functions $p(x_0)$ and $p(x_{t+1}|x_t, u_t)$ for any $t \in [\![0, T-1]\!]$[1]. Then, it holds that*

$$p(u_t|x_t, \mathcal{O}_{t:T} = 1) = \exp\left(-\mathsf{Q}_t(x_t, u_t) + \mathsf{V}_t(x_t)\right), \quad \forall x_t \in \mathbb{X}, \forall u_t \in \mathbb{U}, \qquad (3)$$

---

[1]When considering discrete variables $x_t, u_t$, the assumption $\mu_L(\mathbb{U}) < \infty$ is replaced by the finiteness of the set $\mathbb{U}$, and the existence of the densities is not required.

*where*

$$\mathsf{V}_t(x_t) = -\log\left[\int_{\mathbb{U}} \exp(-\mathsf{Q}_t(x_t, u_t))\mathrm{d}u_t\right], \ \forall t \in [\![0, T-1]\!], \ \mathsf{V}_T(x_T) = c_T(x_T), \quad (4)$$

$$\mathsf{Q}_t(x_t, u_t) = c_t(x_t, u_t) - \log \mathbb{E}_{p(x_{t+1}|x_t, u_t)}\left[\exp(-\mathsf{V}_{t+1}(x_{t+1}))\right], \ \forall t \in [\![0, T-1]\!]. \quad (5)$$

$$\diamondsuit$$

The recursive computation (4), (5) is similar to the Bellman equation for the risk-seeking control. However, it is not still clear what kind of performance index the optimal trajectory $p(\tau|\mathcal{O}_{t:T})$ optimizes because (4) does not coincide with that of the conventional risk-seeking control. An indirect way to make this clear is variational inference. Let us consider finding the closest trajectory distribution $p^\pi(\tau)$ to the optimal distribution $p(\tau|\mathcal{O}_{0:T})$. The variational distribution is chosen as

$$p^\pi(\tau) = p(x_0) \prod_{t=0}^{T-1} p(x_{t+1}|x_t, u_t)\pi_t(u_t|x_t), \quad (6)$$

where $\pi_t(\cdot|x_t) \in \mathcal{P}(\mathbb{U})$ is the conditional density of $u_t$ given $x_t$ and corresponds to a control policy. Then, the minimization of the KL divergence $D_1(p^\pi(\tau)\|p(\tau|\mathcal{O}_{0:T}))$ is known to be equivalent to the following MaxEnt control problem:

$$\underset{\{\pi_t\}_{t=0}^{T-1}}{\text{minimize}} \ \mathbb{E}_{p^\pi(\tau)}\left[c_T(x_T) + \sum_{t=0}^{T-1}\left(c_t(x_t, u_t) - \mathcal{H}_1(\pi_t(\cdot|x_t))\right)\right]. \quad (7)$$

Especially when the system $p(x_{t+1}|x_t, u_t)$ is deterministic, the minimum value of $D_1(p^\pi(\tau)\|p(\tau|\mathcal{O}_{0:T}))$ is 0, and the posterior $p(u_t|x_t, \mathcal{O}_{t:T})$ yields the optimal control of (7). As mentioned in Introduction, this work uses Rényi divergence rather than the KL divergence. Moreover, we characterize the optimal posterior $p(u_t|x_t, \mathcal{O}_{t:T})$ more directly even for stochastic systems.

## 3 Control as Rényi divergence variational inference

In this section, we address the question of what kind of control problem is solved by CaI with Rényi divergence and characterize the optimal policy.

### 3.1 Equivalence between CaI with Rényi divergence and risk-sensitive control

Let $\eta > -1, \eta \neq 0$. Then, CaI using Rényi variational inference is formulated as the minimization of $D_{1+\eta}(p^\pi(\tau)\|p(\tau|\mathcal{O}_{0:T}))$ with respect to $p^\pi$ in (6). Now, we have

$$D_{1+\eta}(p^\pi\|p(\cdot|\mathcal{O}_{0:T})) = \underbrace{\frac{1}{\eta}\log\left[\int p^\pi(\tau)^{1+\eta}p(\tau, \mathcal{O}_{0:T})^{-\eta}\mathrm{d}\tau\right]}_{-(\text{Variational Rényi bound})} + \log p(\mathcal{O}_{0:T}). \quad (8)$$

That is, CaI with Rényi divergence is equivalent to maximizing the above variational Rényi bound. Moreover, by (1), it holds that

$$\log\left[\int p^\pi(\tau)^{1+\eta}p(\tau, \mathcal{O}_{0:T})^{-\eta}\mathrm{d}\tau\right]$$

$$= \log\left[\int p^\pi(\tau)\left(\frac{p(x_0)\prod_{t=0}^{T-1}p(x_{t+1}|x_t, u_t)\pi_t(u_t|x_t)}{\frac{1}{\mu_L(\mathbb{U})}p(x_0)\left[\prod_{t=0}^{T-1}p(x_{t+1}|x_t, u_t)\right]\exp\left(-c_T(x_T) - \sum_{t=0}^{T-1}c_t(x_t, u_t)\right)}\right)^\eta\mathrm{d}\tau\right]$$

$$= \log\left[\int p^\pi(\tau)\exp\left(\eta c_T(x_T) + \eta\sum_{t=0}^{T-1}\left(c_t(x_t, u_t) + \log\pi_t(u_t|x_t)\right)\right)\mathrm{d}\tau\right] + \eta\log\mu_L(\mathbb{U}).$$

Consequently, we obtain the first equivalence result in this paper.

**Theorem 2.** *Suppose that the assumptions in Proposition 1 hold. Then, for any $\eta > -1, \eta \neq 0$, the minimization of $D_{1+\eta}(p^\pi\|p(\cdot|\mathcal{O}_{0:T} = 1))$ with respect to $p^\pi$ in (6) is equivalent to*

$$\underset{\{\pi_t\}_{t=0}^{T-1}}{\text{minimize}} \ \frac{1}{\eta}\log\mathbb{E}_{p^\pi(\tau)}\left[\exp\left(\eta c_T(x_T) + \eta\sum_{t=0}^{T-1}\left(c_t(x_t, u_t) + \log\pi_t(u_t|x_t)\right)\right)\right]. \quad (9)$$

$$\diamondsuit$$

Problem (9) is a risk-sensitive control problem with the log-probability regularization $\log \pi_t(u_t|x_t)$ of the control policy. Let $\eta\Phi(\tau)$ be the exponent in (9). Then, $\frac{1}{\eta}\log\mathbb{E}[\exp(\eta\Phi(\tau))] = \mathbb{E}[\Phi(\tau)] + \frac{\eta}{2}\text{Var}[\Phi(\tau)] + O(\eta^2)$, where $\text{Var}[\cdot]$ denotes the variance [42]. Hence, $\eta > 0$ (resp. $\eta < 0$) leads to risk-averse (resp. risk-seeking) policies. As $\eta$ goes to zero, the objective in (9) converges to the risk-neutral MaxEnt control problem (7).

## 3.2 Derivation of optimal control and further equivalence results

In this subsection, we derive the optimal policy of (9) and give its characterizations. For the analysis, we do not need the non-negativity of the cost $c_t$. We only sketch the derivation, and the detailed proof is given in Appendix B. Similar to the conventional optimal control problems, we adopt the dynamic programming. Another approach based on variational inference will be given in Subsection 4.2. Define the optimal (state-)value function $V_t : \mathbb{X} \to \mathbb{R}$ and the Q-function $\mathcal{Q}_t : \mathbb{X} \times \mathbb{U} \to \mathbb{R}$ as follows:

$$V_t(x_t) := \inf_{\{\pi_s\}_{s=t}^{T-1}} \frac{1}{\eta} \log \mathbb{E}_{p^\pi(\tau|x_t)} \left[ \exp\left( \eta c_T(x_T) + \eta \sum_{s=t}^{T-1} \left( c_s(x_s,u_s) + \log \pi_s(u_s|x_s) \right) \right) \right], \tag{10}$$

$$\mathcal{Q}_t(x_t,u_t) := c_t(x_t,u_t) + \frac{1}{\eta} \log \mathbb{E}_{p(x_{t+1}|x_t,u_t)} \left[ \exp\left( \eta V_{t+1}(x_{t+1}) \right) \right], \quad t \in [\![0, T-1]\!], \tag{11}$$

and $V_T(x_T) := c_T(x_T)$. Then, it can be shown that the Bellman equation for Problem (9) is

$$V_t(x_t) = -\log \left[ \int_{\mathbb{U}} \exp\left(-\mathcal{Q}_t(x_t,u')\right) \mathrm{d}u' \right] + \inf_{\pi_t(\cdot|x_t)\in\mathcal{P}(\mathbb{U})} D_{1+\eta}(\pi_t(\cdot|x_t)\|\pi_t^*(\cdot|x_t)), \tag{12}$$

where $\pi_t^*(u_t|x_t) := \exp\left(-\mathcal{Q}_t(x_t,u_t)\right) / \mathcal{Z}_t(x_t)$, and the normalizing constant is assumed to fulfill $\mathcal{Z}_t(x_t) := \int_{\mathbb{U}} \exp\left(-\mathcal{Q}_t(x_t,u')\right) \mathrm{d}u' < \infty$. Since $D_{1+\eta}(\pi_t(\cdot|x_t)\|\pi_t^*(\cdot|x_t))$ attains its minimum value 0 if and only if $\pi_t(\cdot|x_t) = \pi_t^*(\cdot|x_t)$, the unique optimal policy that minimizes the right-hand side of (12) is given by $\pi_t^*(\cdot|x_t)$ and

$$V_t(x_t) = -\log \left[ \int_{\mathbb{U}} \exp\left(-\mathcal{Q}_t(x_t,u')\right) \mathrm{d}u' \right], \quad \pi_t^*(u_t|x_t) = \exp\left(-\mathcal{Q}_t(x_t,u_t) + V_t(x_t)\right). \tag{13}$$

Because of the softmin operation above, the left equation in (13) is called the soft Bellman equation.

**Theorem 3.** *Assume that $\int_{\mathbb{U}} \exp\left(-\mathcal{Q}_t(x,u')\right) \mathrm{d}u' < \infty$ holds for any $t \in [\![0, T-1]\!]$ and $x \in \mathbb{X}$. Let $\eta > -1$, $\eta \neq 0$. Then, the unique optimal policy of Problem (9) is given by (13). Especially when the dynamics is deterministic, i.e., $p(x_{t+1}|x_t,u_t) = \delta(x_{t+1} - \bar{f}_t(x_t,u_t))$ for some $\bar{f}_t : \mathbb{X} \times \mathbb{U} \to \mathbb{X}$ and the Dirac delta function $\delta$, it holds that*

$$\mathcal{Q}_t(x_t,u_t) = c_t(x_t,u_t) + V_{t+1}\left(\bar{f}_t(x_t,u_t)\right), \tag{14}$$

*and the optimal policy of the MaxEnt control problem (7) solves the LP-regularized risk-sensitive control problem (9) for any $\eta > -1$, $\eta \neq 0$.* ◇

Assumption $\int_{\mathbb{U}} \exp\left(-\mathcal{Q}_t(x,u')\right) \mathrm{d}u' < \infty$ is satisfied for example when $c_t$ is bounded for any $t \in [\![0,T]\!]$ and $\mu_L(\mathbb{U}) < \infty$. The linear quadratic setting also fulfills this assumption; see (16).

Theorem 3 suggests several equivalence results:

**RCaI and MaxEnt control for deterministic systems.** First, we emphasize that even though the equivalence between *unregularized* risk-neutral and risk-sensitive controls for deterministic systems is already known, our equivalence result for MaxEnt and regularized risk-sensitive controls is nontrivial. This is because the regularized policy $\pi_t^*$ makes a system stochastic even though the original system is deterministic, and for stochastic systems, the unregularized risk-sensitive control does not coincide with the risk-neutral control. This implies that the optimal randomness introduced by the regularization does not affect the risk sensitivity of the policy. This provides insight into the robustness of MaxEnt control [8]. Note that [43] mentioned that the MaxEnt control objective can be reconstructed by the risk-sensitive control objective under the heuristic assumption that the cost follows a uniform distribution. However, this assumption is not satisfied in general. Our equivalence result does not require such an unrealistic assumption.

**RCaI and optimal posterior.** Although the optimal posterior $p(u_t|x_t, \mathcal{O}_{t:T})$ yields the MaxEnt control for deterministic systems as mentioned in Section 2, it is not known what objective $p(u_t|x_t, \mathcal{O}_{t:T})$

optimizes for stochastic systems. Theorem 3 gives a new characterization of $p(u_t|x_t, \mathcal{O}_{t:T})$. By formally substituting $\eta = -1$ into (11), the Bellman equation for computing $\pi_t^*$ becomes (4), (5) for the optimal posterior $p(u_t|x_t, \mathcal{O}_{t:T})$. Note that even if the cost function $c_t$ in (9) is replaced by $\mathsf{c}_t$ in Proposition 1, $\{\pi_t^*\}$ is still optimal. Therefore, by taking the limit as $\eta \searrow -1$, the policy $\pi_t^*(u_t|x_t)$ in Theorem 3 converges to $p(u_t|x_t, \mathcal{O}_{t:T})$, and in this sense, the policy $p(u_t|x_t, \mathcal{O}_{t:T})$ is risk-*seeking*.

**Corollary 4.** *Under the assumptions in Proposition 1, it holds that*

$$\lim_{\eta \searrow -1} \pi_t^*(u_t|x_t) = \exp(-\mathcal{Q}_t(x_t, u_t) + V_t(x_t)) = p(u_t|x_t, \mathcal{O}_{t:T} = 1), \tag{15}$$

*where $V_t$ and $\mathcal{Q}_t$ are given by* (11), (13) *with $\eta = -1$.* $\diamond$

**RCaI for deterministic systems and linearly-solvable control.** For deterministic systems, by the transformation $E_t(x_t) := \exp(-V_t(x_t))$, the Bellman equation (14) becomes linear: $E_t(x_t) = \int \exp(-c_t(x_t, u'))E_{t+1}(\bar{f}_t(x_t, u'))\mathrm{d}u'$. That is, when the system is deterministic, the LP-regularized risk-sensitive control, or equivalently, the MaxEnt control is linearly solvable [15, 16, 44], which enables efficient computation of RL. Even for the MaxEnt control, this fact seems not to be mentioned explicitly in the literature.

**RCaI and unregularized risk-sensitive control in linear quadratic setting.** Similar to the unregularized and MaxEnt problems [45, 46], Problem (9) with a linear system $p(x_{t+1}|x_t, u_t) = \mathcal{N}(x_{t+1}|A_t x_t + B_t u_t, \Sigma_t)$ and quadratic costs $c_t(x_t, u_t) = (x_t^\top Q_t x_t + u_t^\top R_t u_t)/2$, $c_T(x_T) = x_T^\top Q_T x_T/2$ admits an explicit form of the optimal policy:

$$\pi_t^*(u|x) = \mathcal{N}\Big(u| - (R_t + B_t^\top \Pi_{t+1}(I - \eta\Sigma_t\Pi_{t+1})^{-1}B_t)^{-1}B_t^\top \Pi_{t+1}(I - \eta\Sigma_t\Pi_{t+1})^{-1}A_t x,$$
$$(R_t + B_t\Pi_{t+1}(I - \eta\Sigma_t\Pi_{t+1})^{-1}B_t)^{-1}\Big). \tag{16}$$

Here, $\mathcal{N}(\cdot|\mu, \Sigma)$ denotes the Gaussian density with mean $\mu$ and covariance $\Sigma$. The definition of $\Pi_t$ and the proof are given in Appendix C. In general, the mean of the regularized risk-sensitive control deviates from the unregularized risk-sensitive control. However, in the linear quadratic Gaussian (LQG) case, the mean of the optimal policy (16) coincides with the optimal control of risk-sensitive LQG control without the regularization [47].

### 3.3 Another risk-sensitive generalization of MaxEnt control via Rényi entropy

The Shannon entropy regularization $\mathbb{E}[-\mathcal{H}_1(\pi_t(\cdot|x_t))]$ of the MaxEnt control problem (7) can be rewritten as $\mathbb{E}[\log \pi_t(u_t|x_t)]$. In this sense, the risk-sensitive control (9) is a natural extension of (7). Nevertheless, for the risk-sensitive case, the interpretation of $\log \pi_t(u_t|x_t)$ as entropy is no longer available. In this subsection, we provide another risk-sensitive extension of the MaxEnt control. Inspired by the Rényi divergence utilized so far, we employ Rényi entropy regularization:

$$\underset{\{\pi_t\}_{t=0}^{T-1}}{\text{minimize}} \ \frac{1}{\eta} \log \mathbb{E}_{p^\pi(\tau)}\left[\exp\left(\eta c_T(x_T) + \eta \sum_{t=0}^{T-1}\Big(c_t(x_t, u_t) - \mathcal{H}_{1-\eta}(\pi_t(\cdot|x_t))\Big)\right)\right], \tag{17}$$

where $\eta \in \mathbb{R} \setminus \{0, 1\}$, and $\pi_t(\cdot|x) \in L^{1-\eta}(\mathbb{U}) := \{\rho \in \mathcal{P}(\mathbb{U})| \int_\mathbb{U} \rho(u)^{1-\eta}\mathrm{d}u < \infty\}, \forall x$, which implies $|\mathcal{H}_{1-\eta}(\pi_t(\cdot|x_t))| < \infty$. As $\eta$ tends to zero, (17) converges to the MaxEnt control problem (7).

Define the value function $\mathcal{V}_t$ and the Q-function $\mathcal{Q}_t$ associated with (17) like (10) and (11). Then, as in Subsection 3.2, the following Bellman equation holds. The derivation is given in Appendix E.

$$\mathcal{V}_t(x_t) = \inf_{\pi_t \in L^{1-\eta}(\mathbb{U})}\left\{\frac{1}{\eta} \log\left[\int_\mathbb{U} \pi_t(u'|x_t)\exp(\eta\mathcal{Q}_t(x_t, u'))\mathrm{d}u'\right] - \mathcal{H}_{1-\eta}(\pi_t(\cdot|x_t))\right\}. \tag{18}$$

For the minimization in (18), we establish the duality between exponential integrals and Rényi entropy like in [40] because the same procedure as for (12) cannot be applied.

**Lemma 5 (Informal).** *For $\beta, \gamma \in \mathbb{R} \setminus \{0\}$ such that $\beta < \gamma$ and for $g : \mathbb{U} \to \mathbb{R}$, it holds that*

$$\frac{1}{\beta} \log\left[\int_\mathbb{U} \exp(\beta g(u))\mathrm{d}u\right] = \inf_{\rho \in L^{1-\frac{\gamma}{\gamma-\beta}}(\mathbb{U})}\left\{\frac{1}{\gamma} \log\left[\int_\mathbb{U} \exp(\gamma g(u))\rho(u)\mathrm{d}u\right] - \frac{1}{\gamma-\beta}\mathcal{H}_{1-\frac{\gamma}{\gamma-\beta}}(\rho)\right\}, \tag{19}$$

*and the unique optimal solution that minimizes the right-hand side of* (19) *is given by*

$$\rho(u) = \frac{\exp\left(-(\gamma - \beta)g(u)\right)}{\int_{\mathbb{U}} \exp(-(\gamma - \beta)g(u'))\mathrm{d}u'}, \quad \forall u \in \mathbb{U}. \tag{20}$$

$$\diamondsuit$$

For the precise statement and the proof, see Appendix D. By applying Lemma 5 with $\beta = \eta - 1$, $\gamma = \eta$ to (18), we obtain the optimal policy of (17) as follows.

**Theorem 6.** *Assume that $c_t$ is bounded below for any $t \in [\![0, T]\!]$. Assume further that for any $x \in \mathbb{X}$ and $t \in [\![0, T-1]\!]$, it holds that $\int_{\mathbb{U}} \exp\left(-\mathcal{Q}_t(x, u')\right)\mathrm{d}u' < \infty$, $\int_{\mathbb{U}} \exp\left(-(1-\eta)\mathcal{Q}_t(x, u')\right)\mathrm{d}u' < \infty$. Then, the unique optimal policy of Problem* (17) *is given by*

$$\pi_t^\star(u_t|x_t) = \frac{1}{\mathscr{Z}_t(x_t)} \exp\left(-\mathcal{Q}_t(x_t, u_t)\right), \quad \forall t \in [\![0, T-1]\!], \ \forall x_t \in \mathbb{X}, \ \forall u_t \in \mathbb{U}, \tag{21}$$

*where $\mathscr{Z}_t(x_t) := \int_{\mathbb{U}} \exp(-\mathcal{Q}_t(x_t, u'))\mathrm{d}u'$, and it holds that*

$$\mathscr{V}_t(x_t) = \frac{-1}{1-\eta} \log\left[\int_{\mathbb{U}} \exp\left(-(1-\eta)\mathcal{Q}_t(x_t, u')\right)\mathrm{d}u'\right], \quad \forall t \in [\![0, T-1]\!], \ \forall x_t \in \mathbb{X}. \tag{22}$$

$$\diamondsuit$$

Recall that the LP regularized risk-sensitive optimal control is given by (11), (13) while the Rényi entropy regularized control is determined by (21), (22), and $\mathcal{Q}_t(x_t, u_t) = c_t(x_t, u_t) + \frac{1}{\eta} \log \mathbb{E}_{p(x_{t+1}|x_t, u_t)}[\exp(\eta \mathscr{V}_{t+1}(x_{t+1}))]$. Hence, the only difference between the risk-sensitive controls for the LP and Rényi regularization is the coefficient in the soft Bellman equations (13), (22).

## 4 Risk-sensitive reinforcement learning via RCaI

Standard RL methods can be derived from CaI using the KL divergence [5]. In this section, we derive risk-sensitive policy gradient and soft actor-critic methods from RCaI.

### 4.1 Risk-sensitive policy gradient

In this subsection, we consider minimizing the cost (9) by a time-invariant policy parameterized as $\pi_t(u|x) = \pi^{(\theta)}(u|x)$, $\theta \in \mathbb{R}^{n_\theta}$. Let $C_\theta(\tau) := c_T(x_T) + \sum_{t=0}^{T-1}(c_t(x_t, u_t) + \log \pi^{(\theta)}(u_t|x_t))$ and $p_\theta$ be the density of the trajectory $\tau$ under the policy $\pi^{(\theta)}$. Then, Problem (9) can be reformulated as the minimization of $J(\theta)/\eta$ where $J(\theta) := \int p_\theta(\tau) \exp(\eta C_\theta(\tau))\mathrm{d}\tau$. To optimize $J(\theta)/\eta$ by gradient descent, we give the gradient $\nabla_\theta J(\theta)$. The proof is shown in Appendix F.

**Proposition 7.** *Assume the existence of densities $p(x_{t+1}|x_t, u_t)$, $p(x_0)$. Assume further that $\pi^{(\theta)}$ is differentiable in $\theta$, and the derivative and the integral can be interchanged as $\nabla_\theta J(\theta) = \int \nabla_\theta[p_\theta(\tau) \exp(\eta C_\theta(\tau))]\mathrm{d}\tau$. Then, for any function $b : \mathbb{R}^{n_x} \to \mathbb{R}$, it holds that*

$$\nabla_\theta J(\theta) = (\eta + 1)\mathbb{E}_{p_\theta(\tau)}\left[\sum_{t=0}^{T-1} \nabla_\theta \log \pi^{(\theta)}(u_t|x_t)\right.$$
$$\left. \times \left\{\exp\left(\eta c_T(x_T) + \eta \sum_{s=t}^{T-1}\left(c_s(x_s, u_s) + \log \pi^{(\theta)}(u_s|x_s)\right)\right) - b(x_t)\right\}\right]. \tag{23}$$

$$\diamondsuit$$

The function $b$ is referred to as a baseline function, which can be used for reducing the variance of an estimate of $\nabla_\theta J$. The following gradient estimate of $J(\theta)/\eta$ is unbiased:

$$\frac{\eta + 1}{\eta} \sum_{t=0}^{T-1} \nabla_\theta \log \pi^{(\theta)}(u_t|x_t)\left\{\exp\left(\eta c_T(x_T) + \eta \sum_{s=t}^{T-1}\left(c_s(x_s, u_s) + \log \pi^{(\theta)}(u_s|x_s)\right)\right) - b(x_t)\right\}.$$

This is almost the same as risk-sensitive REINFORCE [19] except for the additional term $\log \pi^{(\theta)}(u_s|x_s)$. In the risk-neutral limit $\eta \to 0$, this estimator converges to the MaxEnt policy gradient estimator [5].

## 4.2 Risk-sensitive soft actor-critic

In Subsection 3.2, we used dynamic programming to obtain the optimal policy $\{\pi_t^*\}$. Rather, in this section, we adopt a standard procedure of variational inference [48]. First, we find the optimal factor $\pi_t$ for fixed $\pi_s$, $s \neq t$ as follows. The proof is deferred to Appendix G.

**Proposition 8.** *For $t \in [\![0, T-1]\!]$, let $\pi_s, s \neq t$ be fixed. Let $\eta > -1, \eta \neq 0$. Then, the optimal factor $\pi_t^\bullet := \arg\min_{\pi_t \in \mathcal{P}(\mathbb{U})} D_{1+\eta}(p^\pi \| p(\cdot | \mathcal{O}_{0:T} = 1))$ is given by*

$$
\pi_t^\bullet(u_t|x_t) = \frac{1}{Z_t(x_t)} \left( \mathbb{E}_{p^\pi(x_{t+1:T}, u_{t+1:T-1}|x_t, u_t)} \left[ \left( \frac{\prod_{s=t+1}^{T-1} \pi_s(u_s|x_s)}{p(\mathcal{O}_t|x_T) \prod_{s=t}^{T-1} p(\mathcal{O}_s|x_s, u_s)} \right)^\eta \right] \right)^{-1/\eta},
\tag{24}
$$

*where $Z_t(x_t)$ is the normalizing constant.* $\diamondsuit$

By (24), the optimal factor $\pi_t^\bullet$ is independent of the past factors $\pi_s$, $s \in [\![0, t-1]\!]$. Therefore, the variational Rényi bound in (8) is maximized by optimizing $\pi_t$ in backward order from $t = T-1$ to $t = 0$, which is consistent with the dynamic programming. Associated with (24), we define

$$
V_t^\pi(x_t) := \frac{1}{\eta} \log \mathbb{E}_{p^\pi(x_{t+1:T}, u_{t:T-1}|x_t)} \left[ \left( \frac{\prod_{s=t}^{T-1} \pi_s(u_s|x_s)}{p(\mathcal{O}_t|x_T) \prod_{s=t}^{T-1} p(\mathcal{O}_s|x_s, u_s)} \right)^\eta \right]
$$

$$
= \frac{1}{\eta} \log \mathbb{E}_{p^\pi(x_{t+1:T}, u_{t:T-1}|x_t)} \left[ \exp \left( \eta c_T(x_T) + \eta \sum_{s=t}^{T-1} \left( c_s(x_s, u_s) + \log \pi_s(u_s|x_s) \right) \right) \right], \quad (25)
$$

which is the value function for the policy $\{\pi_s\}_{s=t}^{T-1}$ satisfying the following Bellman equation.

$$
V_t^\pi(x_t) = \frac{1}{\eta} \log \mathbb{E}_{\pi_t(u_t|x_t)} \left[ \left( \frac{\pi_t(u_t|x_t)}{p(\mathcal{O}_t|x_t, u_t)} \right)^\eta \mathbb{E}_{p(x_{t+1}|x_t, u_t)} \left[ \exp(\eta V_{t+1}^\pi(x_{t+1})) \right] \right]
\tag{26}
$$

$$
= \frac{1}{\eta} \log \mathbb{E}_{\pi_t(u_t|x_t)} \left[ \exp \left( \eta c_t(x_t, u_t) + \eta \log \pi_t(u_t|x_t) \right) \mathbb{E}_{p(x_{t+1}|x_t, u_t)} \left[ \exp(\eta V_{t+1}^\pi(x_{t+1})) \right] \right].
$$

By the value function, $\pi_t^\bullet(u_t|x_t)$ can be written as

$$
\pi_t^\bullet(u_t|x_t) = \frac{p(\mathcal{O}_t|x_t, u_t)}{Z_t(x_t)} \mathbb{E}_{p(x_{t+1:T}, u_{t+1:T-1}|x_t, u_t)} \left[ \left( \frac{\prod_{s=t+1}^{T-1} \pi_s(u_s|x_s)}{p(\mathcal{O}_t|x_T) \prod_{s=t+1}^{T-1} p(\mathcal{O}_s|x_s, u_s)} \right)^\eta \right]^{-1/\eta}
$$

$$
= \frac{p(\mathcal{O}_t|x_t, u_t)}{Z_t(x_t)} \mathbb{E}_{p(x_{t+1}|x_t, u_t)} \left[ \exp(\eta V_{t+1}^\pi(x_{t+1})) \right]^{-1/\eta}.
\tag{27}
$$

Next, we define the Q-function for $\{\pi_s\}_{s=t+1}^{T-1}$ as follows:

$$
Q_t^\pi(x_t, u_t) := -\log p(\mathcal{O}_t|x_t, u_t) + \frac{1}{\eta} \log \mathbb{E}_{p(x_{t+1}|x_t, u_t)} \left[ \exp(\eta V_{t+1}^\pi(x_{t+1})) \right].
\tag{28}
$$

Then, it follows from (26) and (27) that

$$
V_t^\pi(x_t) = \frac{1}{\eta} \log \mathbb{E}_{\pi_t(u_t|x_t)} \left[ \pi_t(u_t|x_t)^\eta \exp(\eta Q_t^\pi(x_t, u_t)) \right],
\tag{29}
$$

$$
\pi_t^\bullet(u_t|x_t) = \frac{1}{Z_t(x_t)} \exp(-Q_t^\pi(x_t, u_t)), \quad Z_t(x_t) = \int_{\mathbb{U}} \exp\left(-Q_t^\pi(x_t, u')\right) \mathrm{d}u'.
\tag{30}
$$

Especially when $\pi_t(u_t|x_t) = \pi_t^\bullet(u_t|x_t)$, it holds that $V_t^\pi(x_t) = -\log\left[\int \exp(-Q_t^\pi(x_t, u'))\mathrm{d}u'\right]$, which coincides with the soft Bellman equation in (13). In summary, in order to obtain the optimal factor $\pi_t^\bullet$, it is sufficient to compute $V_t^\pi$ and $Q_t^\pi$ in a backward manner.

Next, we consider the situation when the policy is parameterized as $\pi_t^{(\theta)}(u_t|x_t)$, $\theta \in \mathbb{R}^{n_\theta}$ and there is no parameter $\theta$ that gives the optimal factor $\pi_t^{(\theta)} = \pi_t^\bullet$. To accommodate this situation, we utilize the variational Rényi bound. One can easily see that the maximization of the Rényi bound in (8) with respect to a single factor $\pi_t$ is equivalent to the following problem.

$$
\underset{\pi_t}{\text{minimize}} \quad \frac{1}{\eta} \log \mathbb{E}_{p^\pi(x_t)} \left[ \mathbb{E}_{\pi_t(u_t|x_t)} \left[ \pi_t(u_t|x_t)^\eta \exp(\eta Q_t^\pi(x_t, u_t)) \right] \right].
\tag{31}
$$

This suggests choosing $\theta$ that minimizes (31) whose $\pi_t$ is replaced by $\pi_t^{(\theta)}$. Note that this is further equivalent to

$$\underset{\theta}{\text{minimize}} \ \mathbb{E}_{p^\pi(x_t)}\left[D_{1+\eta}\left(\pi_t^{(\theta)}(\cdot|x_t)\,\Big\|\,\frac{\exp(-Q_t^\pi(x_t,\cdot))}{Z_t(x_t)}\right)\right]. \tag{32}$$

We also parameterize $V_t^\pi$ and $Q_t^\pi$ as $V^{(\psi)}$, $Q^{(\phi)}$ and optimize $\psi, \phi$ so that the relations (28), (29) approximately hold. To obtain unbiased gradient estimators later, we minimize the following squared residual error based on (28), (29), and the transformation $T_\eta(v) := (\mathrm{e}^{\eta v} - 1)/\eta$, $v \in \mathbb{R}$:

$$\mathcal{J}_Q(\phi) := \mathbb{E}_{p^\pi(x_t,u_t)}\left[\frac{1}{2}\left\{T_\eta\left(Q^{(\phi)}(x_t,u_t) - c(x_t,u_t)\right) - \mathbb{E}_{p(x_{t+1}|x_t,u_t)}\left[T_\eta(V^{(\psi)}(x_{t+1}))\right]\right\}^2\right],$$

$$\mathcal{J}_V(\psi) := \mathbb{E}_{p^\pi(x_t)}\left[\frac{1}{2}\left\{T_\eta(V^{(\psi)}(x_t)) - \mathbb{E}_{\pi^{(\theta)}(u_t|x_t)}\left[T_\eta\left(Q^{(\phi)}(x_t,u_t) + \log\pi^{(\theta)}(u_t|x_t)\right)\right]\right\}^2\right].$$

Using $Q^{(\phi)}$ and $T_\eta$, we replace (31) with the following equivalent objective:

$$\mathcal{J}_\pi(\theta) := \mathbb{E}_{p^\pi(x_t)}\left[\mathbb{E}_{\pi^{(\theta)}(u_t|x_t)}\left[T_\eta\left(Q^{(\phi)}(x_t,u_t) + \log\pi^{(\theta)}(u_t|x_t)\right)\right]\right]. \tag{33}$$

Noting that $\lim_{\eta\to 0} T_\eta(\kappa(\eta)) = \kappa(0)$ for $\kappa : \mathbb{R} \to \mathbb{R}$, as the risk sensitivity $\eta$ goes to zero, the objectives $\mathcal{J}_Q, \mathcal{J}_V, \mathcal{J}_\pi$ converge to those used for the risk-neutral soft actor-critic [7]. Now, we have

$$\nabla_\phi \mathcal{J}_Q(\phi) = \mathbb{E}_{p^\pi(x_t,u_t)}\Big[\left(\nabla_\phi Q^{(\phi)}(x_t,u_t)\right)\exp\big(\eta Q^{(\phi)}(x_t,u_t) - \eta c(x_t,u_t)\big)$$
$$\times \left\{T_\eta\big(Q^{(\phi)}(x_t,u_t) - c(x_t,u_t)\big) - \mathbb{E}_{p(x_{t+1}|x_t,u_t)}\big[T_\eta(V^{(\psi)}(x_{t+1}))\big]\right\}\Big], \tag{34}$$

$$\nabla_\psi \mathcal{J}_V(\psi) = \mathbb{E}_{p^\pi(x_t)}\Big[\left(\nabla_\psi V^{(\psi)}(x_t)\right)\exp(\eta V^{(\psi)}(x_t))$$
$$\times \left\{T_\eta(V^{(\psi)}(x_t)) - \mathbb{E}_{\pi^{(\theta)}(u_t|x_t)}\big[T_\eta\big(Q^{(\phi)}(x_t,u_t) + \log\pi^{(\theta)}(u_t|x_t)\big)\big]\right\}\Big], \tag{35}$$

$$\nabla_\theta \mathcal{J}_\pi(\theta) = (\eta + 1)\mathbb{E}_{p^\pi(x_t,u_t)}\Big[\left(\nabla_\theta \log\pi^{(\theta)}(u_t|x_t)\right)T_\eta\big(Q^{(\phi)}(x_t,u_t) + \log\pi^{(\theta)}(u_t|x_t)\big)\Big]. \tag{36}$$

Thanks to the transformation $T_\eta$, the expectations appear linearly, and an unbiased gradient estimator can be obtained by removing them. By simply replacing the gradients of the soft actor-critic [7] with (34)–(36), we obtain the risk-sensitive soft actor-critic (RSAC). It is worth mentioning that since RSAC requires only minor modifications to SAC, techniques for stabilizing SAC, e.g., reparameterization, minibatch sampling with a replay buffer, target networks, double Q-network, can be directly used for RSAC.

## 5 Experiment

Unregularized risk-averse control is known to be robust against perturbations in systems [32]. Since the robustness of the regularized cases has not yet been established theoretically, we verify the robustness of policies learned by RSAC through a numerical example. The environment is Pendulum-v1 in OpenAI Gymnasium. We trained control policies using the hyperparameters shown in Appendix H. There were no significant differences in the control performance obtained or the behavior during training. On the other hand, for each $\eta$, one control policy was selected and was applied to a slightly different environment *without retraining*. To be more precise, the pendulum length $l$, which is 1.0 during training, is changed to 1.25 and 1.5; See Fig. 3. In this example, it can be seen that the control policy obtained with larger $\eta$ has a smaller performance degradation due to environmental changes. This robustness can be considered a benefit of risk-sensitive control.

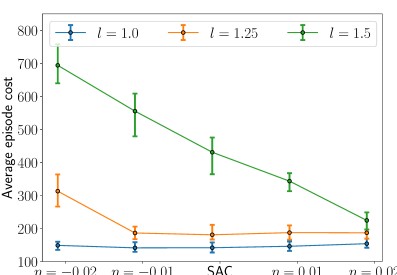

Figure 3: Average episode cost for RSAC with some $\eta$ and standard SAC.

In Fig. 4, empirical distributions of the costs for different risk-sensitivity parameters $\eta$ are plotted. Only the distribution for $\eta = 0.02$ does not change so much under the system perturbations. The

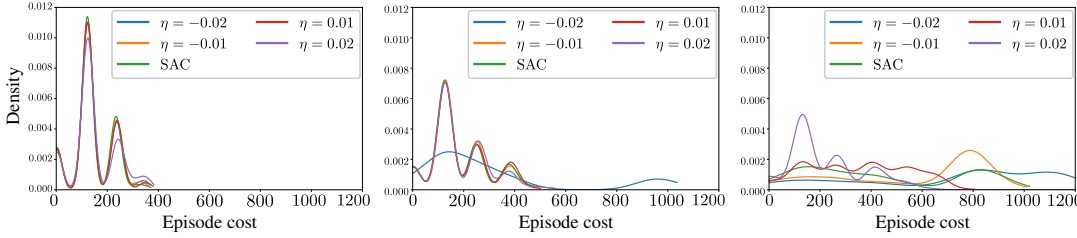

(a) Pendulum length $l = 1.0$ during training

(b) System perturbation $l = 1.25$

(c) System perturbation $l = 1.5$

Figure 4: Empirical distributions of the costs for different risk-sensitivity parameters $\eta$.

distribution for SAC ($\eta = 0$) with $l = 1.5$ deviates from the original one ($l = 1.0$), and another peak of the distribution appears in the high-cost area. This means that there is a high probability of incurring a high cost, which clarifies the advantage of RSAC. The more risk-seeking the policy becomes, the less robust it becomes against the system perturbation.

## 6 Conclusions

In this paper, we proposed a unifying framework of CaI, named RCaI, using Rényi divergence variational inference. We revealed that RCaI yields the LP regularized risk-sensitive control with exponential performance criteria. Moreover, we showed the equivalences for risk-sensitive control, MaxEnt control, the optimal posterior for CaI, and linearly-solvable control. In addition to these connections, we derived the policy gradient method and the soft actor-critic method for the risk-sensitive RL via RCaI. Interestingly, Rényi entropy regularization also results in the same form of the risk-sensitive optimal policy and the soft Bellman equation as the LP regularization.

From a practical point of view, a major limitation of the proposed risk-sensitive soft actor-critic is its numerical instability for large $|\eta|$ cases. Since $\eta$ appears, for example, as $\exp(\eta Q^{(\phi)}(x_t, u_t))$ in the gradients (34)–(36), the magnitude of $\eta$ that does not cause the numerical instability depends on the scale of costs. Therefore, we need to choose $\eta$ depending on environments. In the experiment using Pendulum-v1, $|\eta|$ that is larger than 0.03 results in the failure of learning due to the numerical instability. Although it is an important future work to address this issue, we would like to note that this issue is not specific to our algorithms, but occurs in general risk-sensitive RL with exponential utility. It is also important how to choose a specific value of the order parameter $1 + \eta$ of Rényi divergence. Since we showed that $\eta$ determines the risk sensitivity of the optimal policy, we can follow previous studies on the choice of the sensitivity parameter of the risk-sensitive control without regularization. The properties of the derived algorithms also need to be explored in future work, e.g., the compatibility of a function approximator for RSAC [49].

### Acknowledgments

The authors thank Ran Wang for his valuable help in conducting the experiment. This work was supported in part by JSPS KAKENHI Grant Numbers JP23K19117, JP24K17297, JP21H04875.

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

# A   More details on Control as Inference

In this appendix, we give more details on CaI. As mentioned in (1), the distribution of the state and control input trajectory given optimality variables satisfies

$$p(\tau|\mathcal{O}_{0:T}) \propto p(\tau, \mathcal{O}_{0:T})$$

$$= \left[ p(\mathcal{O}_T|x_T) \prod_{t=0}^{T-1} p(\mathcal{O}_t|x_t, u_t) \right] \left[ p(x_0) \prod_{t=0}^{T-1} p(x_{t+1}|x_t, u_t)p(u_t) \right],$$

where $p(u_t) = 1/\mu_L(\mathbb{U})$ and $p(\tau, \mathcal{O}_{0:T})$ is defined so that

$$\mathbb{P}(\tau \in \mathcal{B}, \ \mathcal{O}_{0:T} = \mathbf{o}_{0:T}) = \int_{\mathcal{B}} p(\tau, \mathbf{o}_{0:T})\mathrm{d}\tau$$

for any $\mathbf{o}_{0:T} \in \{0,1\}^{T+1}$ and any Borel set $\mathcal{B}$, where $\mathbb{P}$ denotes the probability. Therefore, we have

$$p(\tau|\mathcal{O}_{0:T} = 1) \propto \left[ p(x_0) \prod_{t=0}^{T-1} p(x_{t+1}|x_t, u_t) \right] \exp\left( -c_T(x_T) - \sum_{t=0}^{T-1} c_t(x_t, u_t) \right).$$

The posterior $p(u_t|x_t, \mathcal{O}_{t:T} = 1)$ given the optimality condition $\mathcal{O}_{t:T} = 1$ is called the optimal policy. We emphasize that the optimality of $p(u_t|x_t, \mathcal{O}_{t:T} = 1)$ is defined by the condition $\mathcal{O}_{t:T} = 1$ rather than by introducing a cost functional, unlike $\pi^*(u_t|x_t)$ in (13). In the following, we drop $= 1$ for $\mathcal{O}_t$.

The optimal policy can be computed as follows. Define

$$\beta_t(x_t, u_t) := p(\mathcal{O}_{t:T}|x_t, u_t), \tag{37}$$

$$\zeta_t(x_t) := p(\mathcal{O}_{t:T}|x_t). \tag{38}$$

Then, it holds that

$$\zeta_t(x_t) = \int_{\mathbb{U}} p(\mathcal{O}_{t:T}|x_t, u_t)p(u_t|x_t)\mathrm{d}u_t = \int_{\mathbb{U}} \beta_t(x_t, u_t)p(u_t)\mathrm{d}u_t = \frac{1}{\mu_L(\mathbb{U})} \int_{\mathbb{U}} \beta_t(x_t, u_t)\mathrm{d}u_t. \tag{39}$$

In addition, we have

$$\beta_t(x_t, u_t) = p(\mathcal{O}_{t:T}|x_t, u_t) = p(\mathcal{O}_t|x_t, u_t)p(\mathcal{O}_{t+1:T}|x_t, u_t)$$

$$= p(\mathcal{O}_t|x_t, u_t) \int_{\mathbb{X}} p(\mathcal{O}_{t+1:T}|x_{t+1})p(x_{t+1}|x_t, u_t)\mathrm{d}x_{t+1}$$

$$= p(\mathcal{O}_t|x_t, u_t) \int_{\mathbb{X}} \zeta_{t+1}(x_{t+1})p(x_{t+1}|x_t, u_t)\mathrm{d}x_{t+1}, \tag{40}$$

$$\zeta_T(x_T) = p(\mathcal{O}_T|x_T) = \exp(-c_T(x_T)),$$

where we used

$$p(\mathcal{O}_{t+1:T}|x_t, u_t) = \int_{\mathbb{X}} p(\mathcal{O}_{t+1:T}, x_{t+1}|x_t, u_t)\mathrm{d}x_{t+1}$$

$$= \int_{\mathbb{X}} p(\mathcal{O}_{t+1:T}|x_{t+1}, x_t, u_t)p(x_{t+1}|x_t, u_t)\mathrm{d}x_{t+1}$$

$$= \int_{\mathbb{X}} p(\mathcal{O}_{t+1:T}|x_{t+1})p(x_{t+1}|x_t, u_t)\mathrm{d}x_{t+1}.$$

In terms of $\beta_t$ and $\zeta_t$, the optimal policy can be written as

$$p(u_t|x_t, \mathcal{O}_{t:T}) = \frac{p(x_t, u_t, \mathcal{O}_{t:T})}{p(x_t, \mathcal{O}_{t:T})}$$

$$= \frac{p(\mathcal{O}_{t:T}|x_t, u_t)}{p(\mathcal{O}_{t:T}|x_t)}p(u_t|x_t)$$

$$= \frac{\beta_t(x_t, u_t)}{\mu_L(\mathbb{U})\zeta_t(x_t)}. \tag{41}$$

Next, by the logarithmic transformation, we define

$$\mathsf{Q}_t(x_t, u_t) := -\log \frac{\beta_t(x_t, u_t)}{\mu_L(\mathbb{U})}, \tag{42}$$

$$\mathsf{V}_t(x_t) := -\log \zeta_t(x_t). \tag{43}$$

Then, by (41), the optimal policy satisfies

$$p(u_t|x_t, \mathcal{O}_{t:T}) = \exp\left(-\mathsf{Q}_t(x_t, u_t) + \mathsf{V}_t(x_t)\right). \tag{44}$$

By (39), it holds that

$$\mathsf{V}_t(x_t) = -\log\left[\int_{\mathbb{U}} \exp(-\mathsf{Q}_t(x_t, u_t))\mathrm{d}u_t\right]. \tag{45}$$

By using (40), we obtain

$$\exp(-\mathsf{Q}_t(x_t, u_t))\mu_L(\mathbb{U}) = \exp(-c_t(x_t, u_t)) \int_{\mathbb{X}} \zeta_{t+1}(x_{t+1})p(x_{t+1}|x_t, u_t)\mathrm{d}x_{t+1},$$

which yields

$$\mathsf{Q}_t(x_t, u_t) = \mathsf{c}_t(x_t, u_t) - \log \mathbb{E}_{p(x_{t+1}|x_t, u_t)}\left[\exp(-\mathsf{V}_{t+1}(x_{t+1}))\right]. \tag{46}$$

Here, we defined $\mathsf{c}_t(x_t, u_t) := c_t(x_t, u_t) + \log \mu_L(\mathbb{U})$. In summary, Proposition 1 holds.

## B   Proof of Theorem 3

This appendix is devoted to the analysis of the following problem:

$$\underset{\{\pi_t\}_{t=0}^{T-1}}{\text{minimize}} \quad \frac{1}{\eta}\log \mathbb{E}\left[\exp\left(\eta c_T(x_T) + \eta \sum_{t=0}^{T-1}\left(c_t(x_t, u_t) + \varepsilon \log \pi_t(u_t|x_t)\right)\right)\right], \tag{47}$$

$$\text{subject to} \quad x_{t+1} = f_t(x_t, u_t, w_t), \ u_t \in \mathbb{U}, \ \forall t \in [\![0, T-1]\!], \tag{48}$$

$$u_t \sim \pi_t(\cdot|x) \text{ given } x_t = x, \tag{49}$$

$$x_0 \sim \mathbb{P}_{x_0}. \tag{50}$$

Here, $\{w_t\}_{t=0}^{T-1}$ is an independent sequence, $x_0$ is independent of $\{w_t\}$, $\varepsilon > 0$ is the regularization parameter, and $\eta$ is the risk-sensitivity parameter satisfying $\eta > -\varepsilon^{-1}$, $\eta \neq 0$. Note that we do not assume the existence of densities $p(x_{t+1}|x_t, u_t)$, $p(x_0)$. To perform dynamic programming for Problem (47), define the value function and the Q-function as

$$V_t(x) := \inf_{\{\pi_s\}_{s=t}^{T-1}} \frac{1}{\eta}\log \mathbb{E}\left[\exp\left(\eta c_T(x_T) + \eta \sum_{s=t}^{T-1}\left(c_s(x_s, u_s) + \varepsilon \log \pi_s(u_s|x_s)\right)\right) \Bigg| x_t = x\right].$$
$$t \in [\![0, T-1]\!], \ x \in \mathbb{X}, \tag{51}$$

$$V_T(x) := c_T(x), \ x \in \mathbb{X},$$

$$\mathcal{Q}_t(x, u) := c_t(x, u) + \frac{1}{\eta}\log \mathbb{E}\left[\exp\left(\eta V_{t+1}(f_t(x, u, w_t))\right)\right], \ t \in [\![0, T-1]\!], \ x \in \mathbb{X}, \ u \in \mathbb{U}.$$
$$\tag{52}$$

Then, under the assumption that $\int_{\mathbb{U}} \exp\left(-\frac{\mathcal{Q}_t(x, u')}{\varepsilon}\right) \mathrm{d}u' < \infty$, we prove that the unique optimal policy of Problem (47) is given by

$$\pi_t^*(u|x) := \frac{\exp\left(-\frac{\mathcal{Q}_t(x, u)}{\varepsilon}\right)}{\int_{\mathbb{U}} \exp\left(-\frac{\mathcal{Q}_t(x, u')}{\varepsilon}\right) \mathrm{d}u'}, \ t \in [\![0, T-1]\!], \ u \in \mathbb{U}, \ x \in \mathbb{X}. \tag{53}$$

First, by definition, we have

$$V_t(x) = \inf_{\{\pi_s\}_{s=t}^{T-1}} \frac{1}{\eta} \log \left[ \int_{\mathbb{U}} \pi_t(u|x) \mathbb{E} \left[ \exp \left( \eta c_t(x,u) + \varepsilon\eta \log \pi_t(u|x) + \eta c_T(x_T) \right. \right. \right.$$

$$\left. \left. \left. + \eta \sum_{s=t+1}^{T-1} \left( c_s(x_s, u_s) + \varepsilon \log \pi_s(u_s|x_s) \right) \right) \, \middle| \, x_t = x, u_t = u \right] \mathrm{d}u \right]$$

$$= \inf_{\{\pi_s\}_{s=t}^{T-1}} \frac{1}{\eta} \log \left[ \int_{\mathbb{U}} \pi_t(u|x) \exp \left( \eta c_t(x,u) + \varepsilon\eta \log \pi_t(u|x) \right) \right.$$

$$\left. \times \mathbb{E} \left[ \exp \left( \eta c_T(x_T) + \eta \sum_{s=t+1}^{T-1} \left( c_s(x_s, u_s) + \varepsilon \log \pi_s(u_s|x_s) \right) \right) \, \middle| \, x_t = x, u_t = u \right] \mathrm{d}u \right]$$

$$= \inf_{\pi_t} \frac{1}{\eta} \log \left[ \int_{\mathbb{U}} \pi_t(u|x) \exp \left( \eta c_t(x,u) + \varepsilon\eta \log \pi_t(u|x) \right) \mathbb{E} \left[ \exp \left( \eta V_{t+1}(f_t(x,u,w_t)) \right) \right] \mathrm{d}u \right].$$

By the definition of the Q-function (52), we get

$$V_t(x) = \inf_{\pi_t(\cdot|x) \in \mathcal{P}(\mathbb{U})} \frac{1}{\eta} \log \left[ \int_{\mathbb{U}} \pi_t(u|x) \exp(\varepsilon\eta \log \pi_t(u|x)) \exp(\eta \mathcal{Q}_t(x,u)) \mathrm{d}u \right]$$

$$= \inf_{\pi_t(\cdot|x) \in \mathcal{P}(\mathbb{U})} \frac{1}{\eta} \log \left[ \int_{\mathbb{U}} \left( \pi_t(u|x) \right)^{1+\varepsilon\eta} \left( \exp \left( \frac{-\mathcal{Q}_t(x,u)}{\varepsilon} \right) \right)^{-\varepsilon\eta} \mathrm{d}u \right]$$

$$= \inf_{\pi_t(\cdot|x) \in \mathcal{P}(\mathbb{U})} \frac{1}{\eta} \log \left[ \left( \int_{\mathbb{U}} \exp \left( \frac{-\mathcal{Q}_t(x,u')}{\varepsilon} \right) \mathrm{d}u' \right)^{-\varepsilon\eta} \int_{\mathbb{U}} \pi_t(u|x)^{1+\varepsilon\eta} \pi_t^*(u|x)^{-\varepsilon\eta} \mathrm{d}u \right]$$

$$= -\varepsilon \log \left[ \int_{\mathbb{U}} \exp \left( -\frac{\mathcal{Q}_t(x,u')}{\varepsilon} \right) \mathrm{d}u' \right] + \inf_{\pi_t(\cdot|x) \in \mathcal{P}(\mathbb{U})} \varepsilon D_{1+\varepsilon\eta}(\pi_t(\cdot|x) \| \pi_t^*(\cdot|x)).$$

Since $D_{1+\varepsilon\eta}(\pi_t(\cdot|x) \| \pi_t^*(\cdot|x))$ attains its minimum value 0 if and only if $\pi_t(\cdot|x) = \pi_t^*(\cdot|x)$, we conclude that

$$V_t(x) = -\varepsilon \log \left[ \int_{\mathbb{U}} \exp \left( -\frac{\mathcal{Q}_t(x,u')}{\varepsilon} \right) \mathrm{d}u' \right], \quad \forall x \in \mathbb{X}, \tag{54}$$

and the unique optimal policy of Problem (47) is given by (53). Moreover, $\pi_t^*$ can be rewritten as

$$\pi_t^*(u|x) = \exp \left( -\frac{\mathcal{Q}_t(x,u)}{\varepsilon} + \frac{V_t(x)}{\varepsilon} \right), \quad t \in [\![0, T-1]\!], \ u \in \mathbb{U}, \ x \in \mathbb{X}. \tag{55}$$

When considering the deterministic system $x_{t+1} = \bar{f}_t(x_t, u_t)$, we immediately obtain the relation

$$\mathcal{Q}_t(x,u) = c_t(x,u) + V_{t+1}(\bar{f}_t(x,u)). \tag{56}$$

On the other hand, the unique optimal policy of the MaxEnt control problem:

$$\operatorname*{minimize}_{\{\pi_t\}_{t=0}^{T-1}} \mathbb{E} \left[ c_T(x_T) + \sum_{t=0}^{T-1} \left( c_t(x_t, u_t) - \varepsilon \mathcal{H}_1(\pi_t(\cdot|x_t)) \right) \right] \tag{57}$$

is also given by (55) whose Q-function (52) is replaced by

$$\mathcal{Q}_t(x,u) = c_t(x,u) + \mathbb{E}[V_{t+1}(f_t(x,u,w_t))].$$

Therefore, when the system is deterministic, the Q-function of the LP regularized risk-sensitive control problem (47) coincides with that of the MaxEnt control problem (57). Consequently, the optimal policy of Problem (57) solves Problem (47) for any $\eta > -\varepsilon^{-1}$, $\eta \neq 0$ for deterministic systems.

## C Linear quadratic Gaussian setting

In this appendix, we derive the regularized risk-sensitive optimal policy in the linear quadratic Gaussian setting.

**Theorem 9.** *Let $p(x_{t+1}|x_t, u_t) = \mathcal{N}(A_t x_t + B_t u_t, \Sigma_t)$ and $c_t(x_t, u_t) = (x_t^\top Q_t x_t + u_t^\top R_t u_t)/2$, $c_T(x_T) = x_T^\top Q_T x_T/2$, where $\Sigma_t$, $Q_t$, and $R_t$ are positive definite matrices for any $t$, and $\mathcal{N}(\mu, \Sigma)$ denotes the Gaussian distribution with mean $\mu$ and covariance $\Sigma$. Let $\mathbb{X} = \mathbb{R}^{n_x}$, $\mathbb{U} = \mathbb{R}^{n_u}$. Assume that there exists a solution $\{\Pi_t\}_{t=0}^T$ to the following Riccati difference equation:*

$$\Pi_t = Q_t + A_t^\top \Pi_{t+1}(I - \eta\Sigma_t\Pi_{t+1} + B_t R_t^{-1} B_t^\top \Pi_{t+1})^{-1} A_t, \ \ \forall t \in [\![0, T-1]\!], \quad (58)$$

$$\Pi_T = Q_T, \quad (59)$$

*such that $\Sigma_t^{-1} - \eta\Pi_{t+1}$ is positive definite for any $t \in [\![0, T-1]\!]$. Here, $I$ denotes the identity matrix of appropriate dimension. Then, the unique optimal policy of Problem* (9) *is given by*

$$\pi_t^*(u|x) = \mathcal{N}\Big(u| - (R_t + B_t^\top \Pi_{t+1}(I - \eta\Sigma_t\Pi_{t+1})^{-1} B_t)^{-1} B_t^\top \Pi_{t+1}(I - \eta\Sigma_t\Pi_{t+1})^{-1} A_t x,$$

$$(R_t + B_t\Pi_{t+1}(I - \eta\Sigma_t\Pi_{t+1})^{-1} B_t)^{-1}\Big). \quad (60)$$

$$\diamondsuit$$

*Proof.* In this proof, for notational simplicity, we often drop the time index $t$ as $A, B$. First, for $t = T - 1$, the Q-function in (11) is

$$\mathcal{Q}_{T-1}(x, u) = \frac{1}{2}\|x\|_{Q_{T-1}}^2 + \frac{1}{2}\|u\|_{R_{T-1}}^2 + \frac{1}{\eta}\log\mathbb{E}\left[\exp\left(\frac{\eta}{2}\|A_{T-1}x + B_{T-1}u + w_{T-1}\|_{\Pi_T}^2\right)\right], \quad (61)$$

where $\|x\|_P^2 := x^\top P x$ for a symmetric matrix $P$. Here, we have

$$\mathbb{E}\left[\exp\left(\frac{\eta}{2}\|Ax + Bu + w_{T-1}\|_{\Pi_T}^2\right)\right]$$

$$= \frac{1}{\sqrt{(2\pi)^{n_x}|\Sigma_{T-1}|}}\int_{\mathbb{R}^{n_x}} \exp\left(-\frac{1}{2}\|w\|_{\Sigma_{T-1}^{-1}}^2 + \frac{\eta}{2}\|Ax + Bu + w\|_{\Pi_T}^2\right)\mathrm{d}w, \quad (62)$$

where $|\Sigma_{T-1}|$ denotes the determinant of $\Sigma_{T-1}$, and

$$-\frac{1}{2}\|w\|_{\Sigma_{T-1}^{-1}}^2 + \frac{\eta}{2}\|Ax + Bu + w\|_{\Pi_T}^2$$

$$= -\frac{1}{2}\left(\|w\|_{\Sigma^{-1}-\eta\Pi}^2 - 2\eta w^\top\Pi(Ax + Bu) - \|Ax + Bu\|_{\eta\Pi}^2\right).$$

By the assumption that $\Sigma_{T-1}^{-1} - \eta\Pi_T$ is positive definite and a completion of squares argument,

$$-\frac{1}{2}\|w\|_{\Sigma_{T-1}^{-1}}^2 + \frac{\eta}{2}\|Ax + Bu + w\|_{\Pi_T}^2$$

$$= -\frac{1}{2}\left(\|w - (\Sigma^{-1} - \eta\Pi)^{-1}\eta\Pi(Ax + Bu)\|_{\Sigma^{-1}-\eta\Pi}^2 - \|\eta\Pi(Ax + Bu)\|_{(\Sigma^{-1}-\eta\Pi)^{-1}}^2 - \|Ax + Bu\|_{\eta\Pi}^2\right).$$

Thus, we obtain

$$\int_{\mathbb{R}^{n_x}} \exp\left(-\frac{1}{2}\|w\|_{\Sigma_{T-1}^{-1}}^2 + \frac{\eta}{2}\|Ax + Bu + w\|_{\Pi_T}^2\right)\mathrm{d}w$$

$$= \sqrt{(2\pi)^{n_x}|(\Sigma^{-1} - \eta\Pi)^{-1}|}\exp\left(\frac{1}{2}\|\eta\Pi(Ax + Bu)\|_{(\Sigma^{-1}-\eta\Pi)^{-1}}^2 + \frac{1}{2}\|Ax + Bu\|_{\eta\Pi}^2\right). \quad (63)$$

Consequently, by (61)–(63), the Q-function can be written as

$$\mathcal{Q}_{T-1}(x, u) = \frac{1}{2}\|x\|_{Q_{T-1}}^2 + \frac{1}{2}\|u\|_{R_{T-1}}^2 + \frac{1}{2\eta}\|\eta\Pi(A_{T-1}x + B_{T-1}u)\|_{(\Sigma_{T-1}^{-1}-\eta\Pi_T)^{-1}}^2$$

$$+ \frac{1}{2}\|A_{T-1}x + B_{T-1}u\|_\Pi^2 + C_{\mathcal{Q}_{T-1}}$$

$$= \frac{1}{2}\|x\|_Q^2 + \frac{1}{2}\|u\|_R^2 + \frac{1}{2}\|Ax + Bu\|_{\eta\Pi(\Sigma^{-1}-\eta\Pi)^{-1}\Pi+\Pi}^2 + C_{\mathcal{Q}_{T-1}}$$

$$= \frac{1}{2}\|x\|_Q^2 + \frac{1}{2}\|u\|_R^2 + \frac{1}{2}\|Ax + Bu\|_{\Pi(I-\eta\Sigma\Pi)^{-1}}^2 + C_{\mathcal{Q}_{T-1}},$$

where the constant $C_{\mathcal{Q}_{T-1}}$ is independent of $(x, u)$. Now, we adopt a completion of squares argument again:

$$
\begin{aligned}
\mathcal{Q}_{T-1}(x, u) &= \frac{1}{2} \Big( \|u\|^2_{R+B^\top \Pi (I-\eta\Sigma\Pi)^{-1} B} + 2x^\top A^\top \Pi (I-\eta\Pi\Sigma)^{-1} B u + \|x\|^2_{Q+A^\top \Pi(I-\eta\Sigma\Pi)^{-1} A} \Big) \\
&\quad + C_{\mathcal{Q}_{T-1}} \\
&= \frac{1}{2} \Big( \|u + (R + B^\top \Pi(I-\eta\Sigma\Pi)^{-1} B)^{-1} B^\top (I - \eta\Pi\Sigma)^{-1} \Pi A x\|^2_{R+B^\top \Pi(I-\eta\Sigma\Pi)^{-1} B} \\
&\quad - \|B^\top (I-\eta\Pi\Sigma)^{-1}\Pi A x\|^2_{(R+B^\top\Pi(I-\eta\Sigma\Pi)^{-1}B)^{-1}} + \|x\|^2_{Q+A^\top\Pi(I-\eta\Sigma\Pi)^{-1}A} \Big) \\
&\quad + C_{\mathcal{Q}_{T-1}} \\
&= \frac{1}{2} \|u + (R + B^\top \Pi_T(I-\eta\Sigma\Pi_T)^{-1}B)^{-1}B^\top\Pi_T(I-\eta\Sigma\Pi_T)^{-1}Ax\|^2_{R+B^\top\Pi_T(I-\eta\Sigma\Pi_T)^{-1}B} \\
&\quad + \frac{1}{2}\|x\|^2_{\Pi_{T-1}} + C_{\mathcal{Q}_{T-1}}.
\end{aligned}
$$

Here, we used $\Pi_T(I - \eta\Sigma_{T-1}\Pi_T)^{-1} = (I - \eta\Pi_T\Sigma_{T-1})^{-1}\Pi_T$ and

$$
\begin{aligned}
\Pi_{T-1} &= Q_{T-1} + A_{T-1}^\top\Pi_T(I - \eta\Sigma_{T-1}\Pi_T + B_{T-1}R_{T-1}^{-1}B_{T-1}^\top\Pi_T)^{-1}A_{T-1} \\
&= Q + A^\top\Pi_T(I-\eta\Sigma_{T-1}\Pi_T)^{-1}A - A^\top\Pi_T(I-\eta\Sigma_{T-1}\Pi_T)^{-1}B \\
&\quad \times (R_{T-1} + B^\top\Pi_T(I-\eta\Sigma_{T-1}\Pi_T)^{-1}B)^{-1}B^\top(I-\eta\Pi_T\Sigma_{T-1})^{-1}\Pi_T A.
\end{aligned}
$$

Therefore, the optimal policy at $t = T-1$ is

$$
\begin{aligned}
\pi_{T-1}^*(u|x) = \mathcal{N}\big(u| &- (R_{T-1} + B^\top\Pi_T(I-\eta\Sigma_{T-1}\Pi_T)^{-1}B)^{-1}B^\top\Pi_T(I-\eta\Sigma_{T-1}\Pi_T)^{-1}Ax, \\
&(R_{T-1} + B^\top\Pi_T(I-\eta\Sigma_{T-1}\Pi_T)^{-1}B)^{-1}\big). \tag{64}
\end{aligned}
$$

The value function is given by

$$
V_{T-1}(x) = -\log\left[\int_{\mathbb{R}^{n_u}} \exp(-\mathcal{Q}_{T-1}(x, u))\mathrm{d}u\right] = \frac{1}{2}\|x\|^2_{\Pi_{T-1}} + C_{V_{T-1}},
$$

where $C_{V_{T-1}}$ does not depend on $x$.

By applying the same argument as above for $t = T-2, \dots, 0$, we arrive at the optimal policy (60) and

$$
V_t(x) = \frac{1}{2}\|x\|^2_{\Pi_t} + C_{V_t}, \tag{65}
$$

$$
\mathcal{Q}_t(x, u)
$$
$$
= \frac{1}{2}\|u + (R_t + B^\top\Pi_{t+1}(I-\eta\Sigma_t\Pi_{t+1})^{-1}B)^{-1}B^\top\Pi_{t+1}(I-\eta\Sigma_t\Pi_{t+1})^{-1}Ax\|^2_{R_t+B^\top\Pi_{t+1}(I-\eta\Sigma_t\Pi_{t+1})^{-1}B}
$$
$$
+ \frac{1}{2}\|x\|^2_{\Pi_t} + C_{\mathcal{Q}_t}, \tag{66}
$$

where $C_{V_t}$ and $C_{\mathcal{Q}_t}$ are independent of $(x, u)$. This completes the proof. $\square$

By the same argument as above, the optimal policy of the Rényi entropy regularized risk-sensitive control problem (17) in the linear quadratic Gaussian setting is also given by (60).

## D  Proof of Lemma 5

First, we give the precise statement of Lemma 5. To this end, for $a, b \in \mathbb{R}$, define

$$
\underline{\mathcal{B}}_{a,b}(\mathbb{U}) := \left\{ g : \mathbb{U} \to \mathbb{R} \,\middle|\, g \text{ is bounded below, } \int_{\mathbb{U}} \exp(ag(u))\mathrm{d}u < \infty, \int_{\mathbb{U}} \exp(bg(u))\mathrm{d}u < \infty \right\}. \tag{67}
$$

Similarly, define $\overline{\mathcal{B}}_{a,b}(\mathbb{U})$ for upper bounded functions. For given $g : \mathbb{U} \to \mathbb{R}$, $a \in \mathbb{R}$, and $\alpha \in \mathbb{R} \setminus \{0, 1\}$, define

$$\mathcal{P}_{a,g}(\mathbb{U}) := \left\{ \rho \in \mathcal{P}(\mathbb{U}) \,\middle|\, \int_{\mathbb{U}} \exp(ag(u))\rho(u)\mathrm{d}u < \infty \right\},$$

$$L^\alpha(\mathbb{U}) := \left\{ \rho \in \mathcal{P}(\mathbb{U}) \,\middle|\, \int_{\mathbb{U}} \rho(u)^\alpha \mathrm{d}u < \infty \right\}.$$

If $\rho \in L^\alpha(\mathbb{U})$ and $\alpha \in (0, 1)$, then it holds that $\mathcal{H}_\alpha(\rho) < \infty$. If $\alpha \in (-\infty, 0) \cap (1, \infty)$, we have $\mathcal{H}_\alpha(\rho) > -\infty$.

Now, we are ready to state the duality lemma.

**Lemma 10.** *For $\beta, \gamma \in \mathbb{R} \setminus \{0\}$ such that $\beta < \gamma$ and for $g \in \underline{\mathcal{B}}_{\{\beta, -(\gamma-\beta)\}}(\mathbb{U})$, it holds that*

$$\frac{1}{\beta} \log \left[ \int_{\mathbb{U}} \exp(\beta g(u))\mathrm{d}u \right] = \inf_{\rho \in L^{1 - \frac{\gamma}{\gamma - \beta}}(\mathbb{U})} \left\{ \frac{1}{\gamma} \log \left[ \int_{\mathbb{U}} \exp(\gamma g(u))\rho(u)\mathrm{d}u \right] - \frac{1}{\gamma - \beta} \mathcal{H}_{1 - \frac{\gamma}{\gamma - \beta}}(\rho) \right\},$$
(68)

*and the unique optimal solution that minimizes the right-hand side of* (68) *is given by*

$$\rho(u) = \frac{\exp\left(-(\gamma - \beta)g(u)\right)}{\int_{\mathbb{U}} \exp(-(\gamma - \beta)g(u'))\mathrm{d}u'}, \quad u \in \mathbb{U}.$$
(69)

*In addition, for $h \in \overline{\mathcal{B}}_{\{\gamma, \gamma - \beta\}}(\mathbb{U})$, it holds that*

$$\frac{1}{\gamma} \log \left[ \int \exp(\gamma h(u))\mathrm{d}u \right] = \sup_{\rho \in L^{\frac{\gamma}{\gamma - \beta}}(\mathbb{U})} \left\{ \frac{1}{\beta} \log \left[ \int \exp(\beta h(u))\rho(u)\mathrm{d}u \right] + \frac{1}{\gamma - \beta} \mathcal{H}_{\frac{\gamma}{\gamma - \beta}}(\rho) \right\},$$
(70)

*and the unique optimal solution that maximizes the right-hand side of* (70) *is given by*

$$\rho(u) = \frac{\exp((\gamma - \beta)h(u))}{\int \exp((\gamma - \beta)h(u'))\mathrm{d}u'}, \quad u \in \mathbb{U}.$$
(71)

$\diamondsuit$

Although the proof is similar to that of the duality between exponential integrals and Rényi divergence [40], it requires more careful analysis because we do not assume the upper boundedness of $g$ and the lower boundedness of $h$, unlike in [40].

*Proof.* For notational simplicity, we often drop $\mathbb{U}$ as $L^\alpha$. First, we note that it is sufficient to prove that for $\alpha > 0, \alpha \neq 1$, $g \in \underline{\mathcal{B}}_{\{\alpha - 1, -1\}}$, and $h \in \overline{\mathcal{B}}_{\{\alpha, 1\}}$, it holds that

$$\frac{1}{\alpha - 1} \log \left[ \int \exp((\alpha - 1)g(u))\mathrm{d}u \right] = \inf_{\rho \in L^{1 - \alpha}} \left\{ \frac{1}{\alpha} \log \left[ \int \exp(\alpha g(u))\rho(u)\mathrm{d}u \right] - \mathcal{H}_{1 - \alpha}(\rho) \right\},$$
(72)

$$\frac{1}{\alpha} \log \left[ \int \exp(\alpha h(u))\mathrm{d}u \right] = \sup_{\rho \in L^\alpha} \left\{ \frac{1}{\alpha - 1} \log \left[ \int \exp((\alpha - 1)h(u))\rho(u)\mathrm{d}u \right] + \mathcal{H}_\alpha(\rho) \right\},$$
(73)

and

$$\rho^*(u) := \frac{\exp(-g(u))}{\int \exp(-g(u'))\mathrm{d}u'}, \quad \rho^{**}(u) := \frac{\exp(h(u))}{\int \exp(h(u'))\mathrm{d}u'}$$
(74)

are the unique optimal solutions to (72), (73), respectively. To see this, note that if (72), (73) hold for $\alpha > 0, \alpha \neq 1$, they hold for any $\alpha \in \mathbb{R} \setminus \{0, 1\}$. Indeed, when $\alpha < 0$, let $\bar{\alpha} := 1 - \alpha > 1$ and for $h \in \overline{\mathcal{B}}_{\{\alpha, 1\}}$, let $\bar{g} := -h$. Since $\bar{g} \in \underline{\mathcal{B}}_{\{\bar{\alpha} - 1, -1\}}$, by (72), we have

$$\frac{1}{\bar{\alpha} - 1} \log \left[ \int \exp((\bar{\alpha} - 1)\bar{g}(u))\mathrm{d}u \right] = \inf_{\rho \in L^{1 - \bar{\alpha}}} \left\{ \frac{1}{\bar{\alpha}} \log \left[ \int \exp(\bar{\alpha}\bar{g}(u))\rho(u)\mathrm{d}u \right] - \mathcal{H}_{1 - \bar{\alpha}}(\rho) \right\}.$$

Therefore, it holds that

$$-\frac{1}{\alpha}\log\left[\int\exp(\alpha h(u))\mathrm{d}u\right] = \inf_{\rho\in L^\alpha}\left\{\frac{1}{1-\alpha}\log\left[\int\exp((\alpha-1)h(u))\rho(u)\mathrm{d}u\right] - \mathcal{H}_\alpha(\rho)\right\}$$

$$= -\sup_{\rho\in L^\alpha}\left\{\frac{1}{\alpha-1}\log\left[\int\exp((\alpha-1)h(u))\rho(u)\mathrm{d}u\right] + \mathcal{H}_\alpha(\rho)\right\},$$

which means that for any $\alpha < 0$ and any $h \in \overline{\mathcal{B}}_{\alpha,1}$, (73) holds. Similarly, by considering $\bar{h} := -g \in \underline{\mathcal{B}}_{\{\bar\alpha,1\}}$ for $g \in \underline{\mathcal{B}}_{\{\alpha-1,-1\}}$, we can see that for any $\alpha < 0$ and any $g \in \underline{\mathcal{B}}_{\{\alpha-1,-1\}}$, (72) holds. Additionally, (72) and (73) with $\alpha = \frac{\gamma}{\gamma-\beta}$, $g = (\gamma-\beta)\widetilde{g}$, $h = (\gamma-\beta)\widetilde{h}$ coincide with (68), (70) where $g$ and $h$ are replaced by $\widetilde{g}, \widetilde{h}$.

In what follows, for $\alpha > 0$, $\alpha \neq 1$, we prove (72). Note that when $\rho \in L^{1-\alpha}$, $|\mathcal{H}_{1-\alpha}(\rho)| < \infty$ holds. Hence, for the minimization of (72), it is sufficient to consider $\rho \in \mathcal{P}_{\alpha,g} \cap L^{1-\alpha}$. The density $\rho^*$ defined in (74) fulfills $\rho^* \in \mathcal{P}_{\alpha,g} \cap L^{1-\alpha}$ because $g \in \underline{\mathcal{B}}_{\{\alpha-1,-1\}}$, and it can be easily seen that

$$\frac{1}{\alpha-1}\log\left[\int\exp((\alpha-1)g(u))\mathrm{d}u\right] = \frac{1}{\alpha}\log\left[\int\exp(\alpha g(u))\rho^*(u)\mathrm{d}u\right] - \mathcal{H}_{1-\alpha}(\rho^*). \quad (75)$$

First, we consider the case $\alpha > 1$. Define $\widetilde{\rho}(u) := \exp((\alpha-1)g(u))$, $\varphi(u) := \exp(-g(u))$. Then, by Hölder's inequality, for any $\rho \in \mathcal{P}_{\alpha,g} \cap L^{1-\alpha}$, it holds that

$$\int\widetilde{\rho}(u)\mathrm{d}u = \int\left(\frac{\varphi(u)}{\rho(u)}\right)^{\frac{\alpha-1}{\alpha}}\left(\frac{\rho(u)}{\varphi(u)}\right)^{\frac{\alpha-1}{\alpha}}\widetilde{\rho}(u)\mathrm{d}u$$

$$\leq \left(\int\left(\frac{\varphi(u)}{\rho(u)}\right)^{\alpha-1}\widetilde{\rho}(u)\mathrm{d}u\right)^{\frac{1}{\alpha}}\left(\int\frac{\rho(u)}{\varphi(u)}\widetilde{\rho}(u)\mathrm{d}u\right)^{\frac{\alpha-1}{\alpha}}$$

$$= \left(\int\rho(u)^{1-\alpha}\mathrm{d}u\right)^{\frac{1}{\alpha}}\left(\int\exp(\alpha g(u))\rho(u)\mathrm{d}u\right)^{\frac{\alpha-1}{\alpha}}. \quad (76)$$

Noting that $\alpha - 1 > 0$ and taking the logarithm of (76), we get for any $\rho \in \mathcal{P}_{\alpha,g} \cap L^{1-\alpha}$,

$$\frac{1}{\alpha-1}\log\left[\int\exp((\alpha-1)g(u))\mathrm{d}u\right] \leq \frac{1}{\alpha}\log\left[\int\exp(\alpha g(u))\rho(u)\mathrm{d}u\right] - \mathcal{H}_{1-\alpha}(\rho).$$

Combining this with (75), the relation (72) holds, and by (75), $\rho^*$ in (74) is an optimal solution. The equality of Hölder's inequality (76) holds if and only if there exist $a_1, a_2 \geq 0$, $a_1 a_2 \neq 0$ such that $a_1\left(\frac{\varphi(u)}{\rho(u)}\right)^{1-\alpha} = a_2\frac{\rho(u)}{\varphi(u)}$ holds $\widetilde{\mu}$-almost everywhere. Here, $\widetilde{\mu}$ is the measure defined by $\widetilde{\rho}$. This condition is satisfied only for $\rho^*$, that is, it is an unique optimal solution.

Next, we analyze the case $\alpha \in (0,1)$. By Hölder's inequality, for any $\rho \in \mathcal{P}_{\alpha,g}$,

$$\int\left(\frac{\varphi(u)}{\rho(u)}\right)^{\alpha-1}\widetilde{\rho}(u)\mathrm{d}u \leq \left(\int 1^{1/\alpha}\widetilde{\rho}(u)\mathrm{d}u\right)^\alpha\left(\int\left[\left(\frac{\varphi(u)}{\rho(u)}\right)^{\alpha-1}\right]^{\frac{1}{1-\alpha}}\widetilde{\rho}(u)\mathrm{d}u\right)^{1-\alpha}$$

$$= \left(\int\widetilde{\rho}(u)\mathrm{d}u\right)^\alpha\left(\int\frac{\rho(u)}{\varphi(u)}\widetilde{\rho}(u)\mathrm{d}u\right)^{1-\alpha},$$

which yields

$$\frac{1}{\alpha-1}\log\left[\int\exp((\alpha-1)g(u))\mathrm{d}u\right] \leq \frac{1}{\alpha}\left[\int\exp(\alpha g(u))\rho(u)\mathrm{d}u\right] - \mathcal{H}_{1-\alpha}(\rho), \quad \forall\rho\in\mathcal{P}_{\alpha,g}. \quad (77)$$

Then, similar to the case $\alpha > 1$, it can be seen that for $\alpha \in (0,1)$, (72) holds and $\rho^*$ is a unique optimal solution.

Next, we show (73) for $\alpha > 1$. Since $\alpha > 1$ and $h$ is upper bounded, it holds that $\rho \in \mathcal{P}_{\alpha-1,h}$. The density $\rho^{**}$ defined in (74) satisfies $\rho^{**} \in \mathcal{P}_{\alpha-1,h} \cap L^\alpha$ because $h \in \mathcal{B}_{\{\alpha,1\}}$, and one can easily see that

$$\frac{1}{\alpha} \log \left[ \int \exp(\alpha h(u)) \mathrm{d}u \right] = \frac{1}{\alpha - 1} \log \left[ \int \exp((\alpha - 1)h(u)) \rho^{**}(u) \mathrm{d}u \right] + \mathcal{H}_\alpha(\rho^{**}).$$

Define $\widehat{\rho}(u) := \exp((\alpha - 1)h(u))\rho(u)$, $\lambda(u) := \exp(-h(u))\rho(u)$. Then, by Hölder's inequality, for any $\rho \in L^\alpha$, it holds that

$$
\begin{aligned}
\int \widehat{\rho}(u)\mathrm{d}u &= \int \lambda(u)^{\frac{\alpha-1}{\alpha}} \lambda(u)^{-\frac{\alpha-1}{\alpha}} \widehat{\rho}(u)\mathrm{d}u \\
&\leq \left( \int \lambda(u)^{\alpha-1} \widehat{\rho}(u)\mathrm{d}u \right)^{\frac{1}{\alpha}} \left( \int \lambda(u)^{-1} \widehat{\rho}(u)\mathrm{d}u \right)^{\frac{\alpha-1}{\alpha}} \\
&= \left( \int \rho(u)^\alpha \mathrm{d}u \right)^{\frac{1}{\alpha}} \left( \int \exp(\alpha h(u))\mathrm{d}u \right)^{\frac{\alpha-1}{\alpha}}.
\end{aligned}
$$

It follows from the above that for any $\rho \in L^\alpha$,

$$\frac{1}{\alpha - 1} \log \left[ \int \exp((\alpha - 1)h(u))\rho(u)\mathrm{d}u \right] \leq \frac{1}{\alpha} \log \left[ \int \exp(\alpha h(u))\mathrm{d}u \right] - \mathcal{H}_\alpha(\rho).$$

Hence, by the same argument as for (72), we can show that (73) holds for $\alpha > 1$, and $\rho^{**}$ is a unique optimal solution.

Lastly, we show (73) for $\alpha \in (0, 1)$. For $\rho \in L^\alpha$, it holds that $|\mathcal{H}_\alpha(\rho)| < \infty$. Then, noting that $\alpha - 1 < 0$, it is sufficient to perform the maximization in (73) for $\rho \in \mathcal{P}_{\alpha-1,h} \cap L^\alpha$. By Hölder's inequality, for any $\rho \in \mathcal{P}_{\alpha-1,h}$, we have

$$
\begin{aligned}
\int \rho^\alpha \mathrm{d}u = \int \lambda(u)^{\alpha-1} \widehat{\rho}(u)\mathrm{d}u &\leq \left( \int 1^{1/\alpha} \widehat{\rho}(u)\mathrm{d}u \right)^\alpha \left( (\lambda(u)^{\alpha-1})^{\frac{1}{1-\alpha}} \widehat{\rho}(u)\mathrm{d}u \right)^{1-\alpha} \\
&= \left( \int \exp((\alpha - 1)h(u))\rho(u)\mathrm{d}u \right)^\alpha \left( \int \exp(\alpha h(u))\mathrm{d}u \right)^{1-\alpha}.
\end{aligned}
$$

Therefore,

$$\frac{1}{\alpha - 1} \log \left[ \int \exp((\alpha - 1)h(u))\rho(u)\mathrm{d}u \right] \leq \frac{1}{\alpha} \log \left[ \int \exp(\alpha h(u))\mathrm{d}u \right] - \mathcal{H}_\alpha(\rho),$$

and similar to the case $\alpha > 1$, we arrive at (73) for $\alpha \in (0, 1)$, and the unique optimal solution is $\rho^{**}$. This completes the proof. $\qquad\square$

## E   Proof of Theorem 6

In this appendix, we analyze the following problem:

$$\underset{\{\pi_t\}_{t=0}^{T-1}}{\text{minimize}} \; \frac{1}{\eta} \log \mathbb{E} \left[ \exp \left( \eta c_T(x_T) + \eta \sum_{t=0}^{T-1} \left( c_t(x_t, u_t) - \varepsilon \mathcal{H}_{1-\varepsilon\eta}(\pi_t(\cdot | x_t)) \right) \right) \right], \qquad (78)$$

where $\varepsilon > 0$, $\eta \in \mathbb{R} \setminus \{0, \varepsilon^{-1}\}$, the system is given by (48)–(50), and $\pi_t(\cdot | x) \in L^{1-\varepsilon\eta}(\mathbb{U}) := \{\rho \in \mathcal{P}(\mathbb{U}) \mid \int_{\mathbb{U}} \rho(u)^{1-\varepsilon\eta} \mathrm{d}u < \infty\}$ for any $x \in \mathbb{X}$ and $t \in [\![0, T-1]\!]$.

Define the value function and the Q-function associated with (78) as

$$\mathcal{V}_t(x) := \underset{\{\pi_s\}_{s=t}^{T-1}}{\inf} \; \frac{1}{\eta} \log \mathbb{E} \left[ \exp \left( \eta c_T(x_T) + \eta \sum_{s=t}^{T-1} \left( c_s(x_s, u_s) - \varepsilon \mathcal{H}_{1-\varepsilon\eta}(\pi_s(\cdot | x_s)) \right) \right) \middle| x_t = x \right],$$
$$t \in [\![0, T-1]\!], \; x \in \mathbb{X}, \qquad (79)$$

$$\mathcal{V}_T(x) := c_T(x), \quad x \in \mathbb{X},$$

$$\mathcal{Q}_t(x, u) := c_t(x, u) + \frac{1}{\eta} \log \mathbb{E} \left[ \exp\left( \eta \mathcal{V}_{t+1}(f_t(x, u, w_t)) \right) \right], \quad t \in [\![0, T-1]\!], \; x \in \mathbb{X}, \; u \in \mathbb{U}.$$
$$(80)$$

For the analysis, we assume the following conditions.

**Assumption 11.** For any $t \in [\![0, T]\!]$, $c_t$ is bounded below. $\diamond$

**Assumption 12.** The Q-function $\mathcal{Q}_t$ in (80) satisfies

$$\int_{\mathbb{U}} \exp\left(-\frac{\mathcal{Q}_t(x,u)}{\varepsilon}\right) \mathrm{d}u < \infty, \quad \int_{\mathbb{U}} \exp\left(-(1-\varepsilon\eta)\frac{\mathcal{Q}_t(x,u)}{\varepsilon}\right) \mathrm{d}u < \infty \tag{81}$$

for any $x \in \mathbb{X}$ and $t \in [\![0, T-1]\!]$. $\diamond$

For example, when $c_t$ is bounded for any $t \in [\![0, T]\!]$, $\mathcal{Q}_t$ is also bounded, and in addition, if $\mu_L(\mathbb{U}) < \infty$, (81) holds. In the linear quadratic setting, Assumption 12 also holds without the boundedness of $c_t$ and $\mathbb{U}$.

Now, we prove Theorem 6 by induction. First, for $t = T - 1$, we have

$$\mathcal{V}_{T-1}(x) = \inf_{\pi_{T-1}(\cdot|x) \in L^{1-\varepsilon\eta}(\mathbb{U})} \left\{ -\varepsilon\mathcal{H}_{1-\varepsilon\eta}(\pi_{T-1}(\cdot|x)) \right.$$
$$\left. + \frac{1}{\eta} \log\left[\int_{\mathbb{U}} \pi_{T-1}(u|x)\mathbb{E}\big[\exp\left(\eta c_{T-1}(x,u) + \eta c_T(x_T)\right) \mid x_{T-1} = x, \ u_{T-1} = u\big] \mathrm{d}u\right] \right\}.$$

The derivation is same as (85) and (86). By the definition of the Q-function in (80), it holds that

$$\mathcal{V}_{T-1}(x) = \inf_{\pi_{T-1}(\cdot|x) \in L^{1-\varepsilon\eta}(\mathbb{U})} \left\{ \frac{1}{\eta} \log\left[\int_{\mathbb{U}} \pi_{T-1}(u|x) \exp(\eta\mathcal{Q}_{T-1}(x,u)) \mathrm{d}u\right] - \varepsilon\mathcal{H}_{1-\varepsilon\eta}(\pi_{T-1}(\cdot|x)) \right\}. \tag{82}$$

Since $c_T$ and $c_{T-1}$ are bounded below, $\mathcal{Q}_{T-1}$ is also bounded below. Therefore, by Assumption 12, $\mathcal{Q}_{T-1}(x, \cdot) \in \underline{\mathcal{B}}_{-(\varepsilon^{-1}-\eta),-\varepsilon^{-1}}(\mathbb{U})$ (see (67) for the definition of $\underline{\mathcal{B}}_{a,b}$), and we can apply Lemma 10 with $\beta = -(\varepsilon^{-1} - \eta)$, $\gamma = \eta$ to (82). As a result,

$$\mathcal{V}_{T-1}(x) = \frac{-1}{\varepsilon^{-1} - \eta} \log\left[\int_{\mathbb{U}} \exp\left(-(\varepsilon^{-1} - \eta)\mathcal{Q}_{T-1}(x,u)\right) \mathrm{d}u\right], \tag{83}$$

and the unique optimal policy that minimizes the right-hand side of (82) is

$$\pi^\star_{T-1}(u|x) = \frac{\exp\left(-\frac{\mathcal{Q}_{T-1}(x,u)}{\varepsilon}\right)}{\int_{\mathbb{U}} \exp\left(-\frac{\mathcal{Q}_{T-1}(x,u')}{\varepsilon}\right) \mathrm{d}u'}. \tag{84}$$

Moreover, since $\mathcal{Q}_{T-1}$ is bounded below, $\mathcal{V}_{T-1}$ is also bounded below.

Next, we assume the induction hypothesis that for some $t \in [\![0, T-2]\!]$, $\{\pi^\star_s\}_{s=t+1}^{T-1}$ is the unique optimal policy of the minimization in the definition of $\mathcal{V}_{t+1}$, and $\mathcal{V}_{t+1}$ is bounded below. By definition,

we have

$$
\begin{aligned}
\mathscr{V}_t(x) &= \inf_{\{\pi_s\}_{s=t}^{T-1}} \frac{1}{\eta} \log \mathbb{E}\Bigg[\exp\bigg(\eta c_t(x, u_t) - \varepsilon\eta\mathcal{H}_{1-\varepsilon\eta}(\pi_t(\cdot|x)) + \eta c_T(x_T) \\
&\qquad\qquad + \eta \sum_{s=t+1}^{T-1} \Big(c_s(x_s, u_s) - \varepsilon\mathcal{H}_{1-\varepsilon\eta}(\pi_s(\cdot|x_s))\Big)\bigg)\bigg|\, x_t = x\Bigg] \\
&= \inf_{\{\pi_s\}_{s=t}^{T-1}} -\varepsilon\mathcal{H}_{1-\varepsilon\eta}(\pi_t(\cdot|x)) \\
&\quad + \frac{1}{\eta} \log \mathbb{E}\Bigg[\exp\bigg(\eta c_t(x, u_t) + \eta c_T(x_T) + \eta \sum_{s=t+1}^{T-1} \Big(c_s(x_s, u_s) - \varepsilon\mathcal{H}_{1-\varepsilon\eta}(\pi_s(\cdot|x_s))\Big)\bigg)\bigg|\, x_t = x\Bigg] \\
&= \inf_{\{\pi_s\}_{s=t}^{T-1}} -\varepsilon\mathcal{H}_{1-\varepsilon\eta}(\pi_t(\cdot|x)) + \frac{1}{\eta} \log\Bigg[\int_{\mathbb{U}} \pi_t(u|x)\mathbb{E}\Bigg[\exp\bigg(\eta c_t(x, u) + \eta c_T(x_T) \\
&\qquad\qquad + \eta \sum_{s=t+1}^{T-1} \Big(c_s(x_s, u_s) - \varepsilon\mathcal{H}_{1-\varepsilon\eta}(\pi_s(\cdot|x_s))\Big)\bigg)\bigg|\, x_t = x,\ u_t = u\Bigg]\mathrm{d}u\Bigg] \\
&= \inf_{\pi_t(\cdot|x)\in L^{1-\varepsilon\eta}(\mathbb{U})} -\varepsilon\mathcal{H}_{1-\varepsilon\eta}(\pi_t(\cdot|x)) + \frac{1}{\eta} \log\Bigg[\int \pi_t(u|x)\exp(\eta c_t(x, u)) \\
&\quad \times \mathbb{E}_{\{\pi_s^\star\}_{s=t+1}^{T-1}}\Bigg[\exp\bigg(\eta c_T(x_T) + \eta \sum_{s=t+1}^{T-1} \Big(c_s(x_s, u_s) - \varepsilon\mathcal{H}_{1-\varepsilon\eta}(\pi_s^\star(\cdot|x_s))\Big)\bigg)\bigg|\, x_t = x,\ u_t = u\Bigg]\mathrm{d}u\Bigg].
\end{aligned}
$$
(85)

Moreover, noting that

$$
\exp(\eta\mathscr{V}_{t+1}(x)) = \mathbb{E}_{\{\pi_s^\star\}_{s=t+1}^{T-1}}\Bigg[\exp\bigg(\eta c_T(x_T) + \eta \sum_{s=t+1}^{T-1} \Big(c_s(x_s, u_s) - \varepsilon\mathcal{H}_{1-\varepsilon\eta}(\pi_s^\star(\cdot|x_s))\Big)\bigg)\bigg|\, x_{t+1} = x\Bigg],
$$

we get

$$
\begin{aligned}
\mathscr{V}_t(x) &= \inf_{\pi_t(\cdot|x)\in L^{1-\varepsilon\eta}(\mathbb{U})} -\varepsilon\mathcal{H}_{1-\varepsilon\eta}(\pi_t(\cdot|x)) \\
&\quad + \frac{1}{\eta} \log\Bigg[\int \pi_t(u|x)\exp(\eta c_t(x, u))\mathbb{E}\big[\exp\big(\eta\mathscr{V}_{t+1}(f_t(x, u, w_t))\big)\big]\,\mathrm{d}u\Bigg].
\end{aligned}
$$
(86)

By using $\mathscr{Q}_t$, the above equation can be written as

$$
\mathscr{V}_t(x) = \inf_{\pi_t(\cdot|x)\in L^{1-\varepsilon\eta}(\mathbb{U})} \frac{1}{\eta} \log\Bigg[\int_{\mathbb{U}} \pi_t(u|x)\exp(\eta\mathscr{Q}_t(x, u))\mathrm{d}u\Bigg] - \varepsilon\mathcal{H}_{1-\varepsilon\eta}(\pi_t(\cdot|x)).
$$
(87)

Since we assumed that $\mathscr{V}_{t+1}$ is bounded below, $\mathscr{Q}_t$ is also bounded below. By combining this with Assumption 12, it holds that $\mathscr{Q}_t(x, \cdot) \in \underline{\mathcal{B}}_{-(\varepsilon^{-1}-\eta),-\varepsilon^{-1}}(\mathbb{U})$. Thus, by Lemma 10, the unique optimal policy that minimizes the right-hand side of the above equation is

$$
\pi_t^\star(u|x) = \frac{\exp\left(-\frac{\mathscr{Q}_t(x, u)}{\varepsilon}\right)}{\int_{\mathbb{U}} \exp\left(-\frac{\mathscr{Q}_t(x, u')}{\varepsilon}\right)\mathrm{d}u'}
$$
(88)

and

$$
\mathscr{V}_t(x) = \frac{-1}{\varepsilon^{-1} - \eta} \log\Bigg[\int_{\mathbb{U}} \exp\left(-(\varepsilon^{-1} - \eta)\mathscr{Q}_t(x, u)\right)\mathrm{d}u\Bigg].
$$
(89)

Lastly, since $\mathscr{Q}_t$ is bounded below, $\mathscr{V}_t$ is also bounded below. This completes the induction step, and we obtain Theorem 6.

## F   Proof of Proposition 7

By using the relation $\nabla_\theta \log p_\theta(\tau) = \nabla_\theta p_\theta(\tau)/p_\theta(\tau)$, we obtain

$$\nabla_\theta J(\theta) = \int p_\theta(\tau) \exp(\eta C_\theta(\tau))\big(\eta \nabla_\theta C_\theta(\tau) + \nabla_\theta \log p_\theta(\tau)\big)\mathrm{d}\tau.$$

In addition, by the expression

$$p_\theta(\tau) = p(x_0) \prod_{t=0}^{T-1} p(x_{t+1}|x_t, u_t)\pi^{(\theta)}(u_t|x_t),$$

we get

$$\nabla_\theta J(\theta) = \int p_\theta(\tau) \exp(\eta C_\theta(\tau)) \left( \eta \sum_{t=0}^{T-1} \nabla_\theta \log \pi^{(\theta)}(u_t|x_t) + \sum_{t=0}^{T-1} \nabla_\theta \log \pi^{(\theta)}(u_t|x_t) \right) \mathrm{d}\tau$$

$$= (\eta+1)\mathbb{E}_{p_\theta(\tau)}\left[ \left( \sum_{t=0}^{T-1} \nabla_\theta \log \pi^{(\theta)}(u_t|x_t) \right) \exp\left( \eta c_T(x_T) + \eta \sum_{t=0}^{T-1} \big(c_t(x_t, u_t) + \log \pi^{(\theta)}(u_t|x_t)\big) \right) \right].$$

$$(90)$$

Note that for any $h : (\mathbb{X})^{t+1} \times (\mathbb{U})^{t+1} \to \mathbb{R}$, it holds that

$$\mathbb{E}\left[h(x_{0:t}, u_{0:t})\right] = \int h(x_{0:t}, u_{0:t})p(x_0) \prod_{s=0}^{T-1} p(x_{s+1}|x_s, u_s)\pi^{(\theta)}(u_s|x_s)\mathrm{d}x_{0:T}\mathrm{d}u_{0:T-1}$$

$$= \int h(x_{0:t}, u_{0:t})p(x_0) \prod_{s=0}^{T-2} p(x_{s+1}|x_s, u_s)\pi^{(\theta)}(u_s|x_s)$$

$$\times \left[ \int p(x_T|x_{T-1}, u_{T-1})\pi^{(\theta)}(u_{T-1}|x_{T-1})\mathrm{d}x_T\mathrm{d}u_{T-1} \right]\mathrm{d}x_{0:T-1}\mathrm{d}u_{0:T-2}$$

$$= \int h(x_{0:t}, u_{0:t})p(x_0) \prod_{s=0}^{T-2} p(x_{s+1}|x_s, u_s)\pi^{(\theta)}(u_s|x_s)\mathrm{d}x_{0:T-1}\mathrm{d}u_{0:T-2}$$

$$\vdots$$

$$= \int h(x_{0:t}, u_{0:t})\pi^{(\theta)}(u_t|x_t)p(x_0) \prod_{s=0}^{t-1} p(x_{s+1}|x_s, u_s)\pi^{(\theta)}(u_s|x_s)\mathrm{d}x_{0:t}\mathrm{d}u_{0:t}.$$

It follows from the above that

$$
\mathbb{E}_{p_\theta(\tau)}\left[\nabla_\theta \log \pi^{(\theta)}(u_t|x_t)\exp\left(\eta\sum_{s=0}^{t-1}\left(c_s(x_s,u_s)+\log\pi^{(\theta)}(u_s|x_s)\right)\right)\right]
$$

$$
= \int \nabla_\theta \log \pi^{(\theta)}(u_t|x_t)\exp\left(\eta\sum_{s=0}^{t-1}\left(c_s(x_s,u_s)+\log\pi^{(\theta)}(u_s|x_s)\right)\right)
$$

$$
\times \pi^{(\theta)}(u_t|x_t)p(x_0)\prod_{s=0}^{t-1}p(x_{s+1}|x_s,u_s)\pi^{(\theta)}(u_s|x_s)\mathrm{d}x_{0:t}\mathrm{d}u_{0:t}
$$

$$
= \int \nabla_\theta \pi^{(\theta)}(u_t|x_t)\exp\left(\eta\sum_{s=0}^{t-1}\left(c_s(x_s,u_s)+\log\pi^{(\theta)}(u_s|x_s)\right)\right)
$$

$$
\times p(x_0)\prod_{s=0}^{t-1}p(x_{s+1}|x_s,u_s)\pi^{(\theta)}(u_s|x_s)\mathrm{d}x_{0:t}\mathrm{d}u_{0:t}
$$

$$
= \int\left(\nabla_\theta\int\pi^{(\theta)}(u_t|x_t)\mathrm{d}u_t\right)\exp\left(\eta\sum_{s=0}^{t-1}\left(c_s(x_s,u_s)+\log\pi^{(\theta)}(u_s|x_s)\right)\right)
$$

$$
\times p(x_0)\prod_{s=0}^{t-1}p(x_{s+1}|x_s,u_s)\pi^{(\theta)}(u_s|x_s)\mathrm{d}x_{0:t}\mathrm{d}u_{0:t-1}
$$

$$
= 0. \tag{91}
$$

By combining this with (90), we get

$$
\nabla_\theta J(\theta)
$$
$$
= (\eta+1)\mathbb{E}_{p_\theta(\tau)}\left[\sum_{t=0}^{T-1}\nabla_\theta\log\pi^{(\theta)}(u_t|x_t)\exp\left(\eta c_T(x_T)+\eta\sum_{s=t}^{T-1}\left(c_s(x_s,u_s)+\log\pi^{(\theta)}(u_s|x_s)\right)\right)\right].
$$
$$
\tag{92}
$$

Lastly, for any function $b:\mathbb{R}^n\to\mathbb{R}$, it holds that

$$
\mathbb{E}_{p_\theta(\tau)}[\nabla_\theta\log\pi^{(\theta)}(u_t|x_t)b(x_t)] = \int p_\theta(x_t,u_t)\frac{\nabla_\theta\pi^{(\theta)}(u_t|x_t)}{\pi^{(\theta)}(u_t|x_t)}b(x_t)\mathrm{d}x_t\mathrm{d}u_t
$$

$$
= \int p(x_t)b(x_t)\nabla_\theta\pi^{(\theta)}(u_t|x_t)\mathrm{d}u_t\mathrm{d}x_t = 0.
$$

This completes the proof.

# G   Proof of Proposition 8

By definition,

$$
\pi_t^\bullet = \underset{\pi_t\in\mathcal{P}(\mathbb{U})}{\arg\min}\frac{1}{\eta}\log\left[\int p^\pi(\tau)\left(\frac{\prod_{s=0}^{T-1}\pi_s(u_s|x_s)}{p(O_T|x_T)\prod_{s=0}^{T-1}p(\mathcal{O}_s|x_s,u_s)}\right)^\eta\mathrm{d}\tau\right]. \tag{93}
$$

The term between the brackets is

$$\int p^\pi(x_{0:t}, u_{0:t})$$

$$\times \left( \int p^\pi(x_{t+1:T}, u_{t+1:T}|x_t, u_t) \left[ \frac{\prod_{s=0}^{T-1} \pi_s(u_s|x_s)}{p(O_T|x_T) \prod_{s=0}^{T-1} p(\mathcal{O}_s|x_s, u_s)} \right]^\eta \mathrm{d}x_{t+1:T}\mathrm{d}u_{t+1:T} \right) \mathrm{d}x_{0:t}\mathrm{d}u_{0:t}$$

$$= \int p^\pi(x_{0:t}, u_{0:t}) \left[ \frac{\prod_{s=0}^{t-1} \pi_s(u_s|x_s)}{\prod_{s=0}^{t-1} p(\mathcal{O}_s|x_s, u_s)} \right]^\eta$$

$$\times \underbrace{\left( \int p^\pi(x_{t+1:T}, u_{t+1:T}|x_t, u_t) \left[ \frac{\prod_{s=t}^{T-1} \pi_s(u_s|x_s)}{p(\mathcal{O}_t|x_T) \prod_{s=t}^{T-1} p(\mathcal{O}_s|x_s, u_s)} \right]^\eta \mathrm{d}x_{t+1:T}\mathrm{d}u_{t+1:T} \right)}_{=:M} \mathrm{d}x_{0:t}\mathrm{d}u_{0:t},$$

where

$$M = \pi_t(u_t|x_t)^\eta \int p^\pi(x_{t+1:T}, u_{t+1:T}|x_t, u_t) \left[ \frac{\prod_{s=t+1}^{T-1} \pi_s(u_s|x_s)}{p(O_T|x_T) \prod_{s=t}^{T-1} p(\mathcal{O}_s|x_s, u_s)} \right]^\eta \mathrm{d}x_{t+1:T}\mathrm{d}u_{t+1:T}.$$

In addition, by the expression $p^\pi(x_{0:t}, u_{0:t}) = p(x_0)\pi_t(u_t|x_t) \prod_{s=0}^{t-1} p(x_{s+1}|x_s, u_s)\pi_s(u_s|x_s)$,

$$\pi_t^\bullet = \arg\min_{\pi_t} \frac{1}{\eta} \log\left[ \int \pi_t(u_t|x_t)^{1+\eta} \right.$$

$$\left. \times \mathbb{E}_{p^\pi(x_{t+1:T}, u_{t+1:T}|x_t, u_t)} \left[ \left( \frac{\prod_{s=t+1}^{T-1} \pi_s(u_s|x_s)}{p(\mathcal{O}_t|x_T) \prod_{s=t}^{T-1} p(\mathcal{O}_s|x_s, u_s)} \right)^\eta \right] \mathrm{d}x_t \mathrm{d}u_t \right]. \quad (94)$$

Now, define

$$\widehat{\pi}_t(u_t|x_t) := \frac{1}{Z_t(x_t)} \left( \mathbb{E}_{p^\pi(x_{t+1:T}, u_{t+1:T}|x_t, u_t)} \left[ \left( \frac{\prod_{s=t+1}^{T-1} \pi_s(u_s|x_s)}{p(\mathcal{O}_t|x_T) \prod_{s=t}^{T-1} p(\mathcal{O}_s|x_s, u_s)} \right)^\eta \right] \right)^{-1/\eta}, \quad (95)$$

$$Z_t(x_t) := \int \left( \mathbb{E}_{p^\pi(x_{t+1:T}, u_{t+1:T}|x_t, u_t)} \left[ \left( \frac{\prod_{s=t+1}^{T-1} \pi_s(u_s|x_s)}{p(\mathcal{O}_t|x_T) \prod_{s=t}^{T-1} p(\mathcal{O}_s|x_s, u_s)} \right)^\eta \right] \right)^{-1/\eta} \mathrm{d}u_t. \quad (96)$$

Then, (94) can be rewritten as

$$\pi_t^\bullet = \arg\min_{\pi_t} \frac{1}{\eta} \log\left[ \int_{\mathbb{X}} Z_t(x_t)^\eta \int_{\mathbb{U}} \widehat{\pi}_t(u_t|x_t) \left( \frac{\pi_t(u_t|x_t)}{\widehat{\pi}_t(u_t|x_t)} \right)^{1+\eta} \mathrm{d}u_t \mathrm{d}x_t \right]. \quad (97)$$

By Jensen's inequality, for any $\eta > -1$, $\eta \neq 0$, it holds that

$$\frac{1}{\eta} \log\left[ \int_{\mathbb{X}} Z_t(x_t)^\eta \int_{\mathbb{U}} \widehat{\pi}_t(u_t|x_t) \left( \frac{\pi_t(u_t|x_t)}{\widehat{\pi}_t(u_t|x_t)} \right)^{1+\eta} \mathrm{d}u_t \mathrm{d}x_t \right]$$

$$\geq \frac{1}{\eta} \log\left[ \int_{\mathbb{X}} Z_t(x_t)^\eta \left( \int_{\mathbb{U}} \widehat{\pi}_t(u_t|x_t) \frac{\pi_t(u_t|x_t)}{\widehat{\pi}_t(u_t|x_t)} \mathrm{d}u_t \right)^{1+\eta} \mathrm{d}x_t \right]$$

$$= \frac{1}{\eta} \log\left[ \int_{\mathbb{X}} Z_t(x_t)^\eta \mathrm{d}x_t \right], \quad (98)$$

where the equality holds if and only if $\pi(\cdot|x_t) \ll \widehat{\pi}_t(\cdot|x_t)$, and $\pi_t(\cdot|x_t)/\widehat{\pi}_t(\cdot|x_t)$ is constant $\widehat{\mathbb{P}}_{x_t}$-almost everywhere. Here, $\widehat{\mathbb{P}}_{x_t}$ is the probability distribution associated with $\widehat{\pi}_t(\cdot|x_t)$. Hence, the infimum (98) is attained only by $\pi_t = \widehat{\pi}_t$. This completes the proof.

## H    Details of the experiment

The implementation of the risk-sensitive SAC (RSAC) algorithm follows the stable-baselines3 [50] version of the SAC algorithm, which means that the RSAC algorithm also implements some tricks including reparameterization, minibatch sampling with a replay buffer, target networks, and double Q-network. Now, we introduce a series of hyperparameters listed in Table 1 shared for both SAC and RSAC algorithms.

Table 1: SAC and RSAC Hyperparameters

| Parameter | Value |
| --- | --- |
| optimizer | Adam [51] |
| learning rate | $10^{-3}$ |
| discount factor | 0.99 |
| regularization coefficient | 0.1 |
| target smoothing coefficient | 0.005 |
| replay buffer size | $10^5$ |
| number of critic networks | 2 |
| number of hidden layers (all networks) | 2 |
| number of hidden units per layer | 256 |
| number of samples per minibatch | 256 |
| activation function | ReLU |

As mentioned in Section 5, there were no significant differences in the control performance obtained or the behavior during training shown in Fig. 5 with those hyperparameters. However, when $\eta$ is too small or too large, the training process becomes unstable due to the gradient vanishing problem and the gradient exponential growth problem, respectively, leading to training failure. To this end, we compare the robustness of the trained policies with RSAC ($\eta \in \{-0.02, -0.01, 0.01, 0.02\}$) and the standard SAC, which corresponds to $\eta = 0$, in the experiment. For each learned policy, we do trail for 20 times. For each trail, we take 100 sampling paths to calculate the average episode cost. In Fig. 3, the error bars depict the max and min values, and the points depict the mean value among the 20 trails. We change the length of the pole $l$ in the Pendulum-v1 environment to test the robustness of the learned policies ($l = 1.0$ m in the original environment). For the training, we used an Ubuntu 20.04 server (GPU: NVIDIA GeForce RTX 2080Ti). The code is available at https://github.com/kaito-1111/risk-sensitive-sac.git.

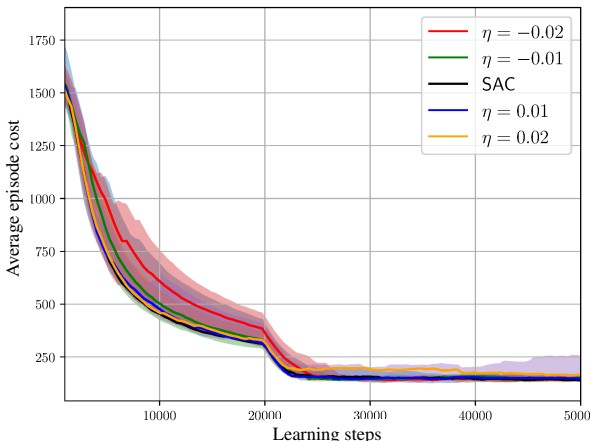

Figure 5: Training process of RSAC (with different $\eta$) and SAC in terms of average episode cost.

