# OpenReview forum: "Risk-sensitive control as inference with Rényi divergence"
_NeurIPS.cc/2024/Conference — NeurIPS 2024 poster_

### Official Review · Reviewer_qM7b · 2024-06-20

**Soundness:** 3
**Presentation:** 3
**Contribution:** 2
**Rating:** 3
**Confidence:** 5

**Summary:**

This paper explores the core research question, "What kind of control problem is solved by control-as-inference (CaI) with Renyi divergence?"  The authors characterize the CaI objective that results from replacing Kullback-Leibler (KL) divergence with the more general Renyi divergence.  They show that the Renyi order parameter $\alpha$ controls risk-sensitivity of the learned policy and refer to the result as risk-sensitive CaI (RCaI).  The paper also proposes a policy gradient method for optimizing the resulting objective via variational inference.  An experiment is provided in the Pendulum-v1 environment from OpenAI Gymnasium.

**Strengths:**

The paper is relatively well-written, easy-to-read, and the ideas discussed are interesting and appear novel.  Risk-sensitive RL / control is an important problem area, and contributions made are likely to have impact.

**Weaknesses:**

While the work is interesting the paper lacks a strong motivation.  The research question of "what kind of problem is solved by CaI with Renyi divergence?" is interesting enough, but the authors propose no hypothesis or motivation for why one would want to consider Renyi divergence.  What is wrong with the existing formulation in terms of KL divergence?  I believe that there is a straightforward answer to this question, but the current manuscript does not address it.  The resulting RSAC algorithm is somewhat more complicated than the SAC counterpart, and so a strong justification for what is gained by considering RSAC in lieu of SAC, or indeed any of the other risk-sensitive RL approaches, would be necessary.

One way to justify the proposed method over existing ones is to empirically demonstrate some advantage.  However, the authors do not provide such empirical comparison.  There is no comparison of the proposed method(s) to baselines from the literature.  The authors only consider a single environment (Pendulum-v1).  Moreover, the notion of "risk" is not clear in the chosen environment.  Overall the experimental validation needs to be more convincing to recommend publication.  The authors should consider more environments, particularly where there is a precise notion of "risk", and compare to baseline methods such as SAC, MBPO, MPO, VMBPO, Mismatched no More, etc.

**Detailed comments**
* L284-286 : It is unclear what the authors intend to demonstrate to show that the proposed method "works." It would be helpful to be more specific.  What are you showing?
* Eq. (23) : The existence of the exponential function in the gradient suggests that learning may be numerically unstable.  The authors should address numerical stability of their approach beyond noting it in the conclusion.
* L147-151 : These are known results and references should be provided for them

**Questions:**

See "Weaknesses" section

**Limitations:**

The authors explicitly note some limitations of the work in the Conclusions section.

---

> ### Author Rebuttal · Authors · 2024-08-07
>
> Thank you very much for your thoughtful comments.
>
> >  While the work is interesting the paper lacks a strong motivation. . . .  risk-sensitive RL approaches, would be necessary.
>
> We apologize for the unclear presentation, which misled the reviewer thinking that our motivation was to improve SAC.
> Our main motivation is to extend the framework of CaI and to provide new theoretical perspectives on CaI.
> As mentioned in the literature review, there has been an attempt to use a divergence other than the KL divergence for CaI, and it was shown that model predictive control based on the Tsallis divergence outperforms the KL divergence in some situations. Despite this advantage found experimentally, no theoretical properties of CaI with divergences other than the KL divergence were known.
> In this work, we have discovered that the Renyi divergence is a natural choice for the extension of CaI in the sense that the resulting policy solves the well-known risk-sensitive control problem with exponential utility.
> In addition to this theoretical discovery, thanks to the proposed unifying framework, we have revealed several equivalences between RCaI, MaxEnt control, the optimal posterior for CaI, and linearly-solvable control.
> In summary, our main contribution is to establish a unifying framework of CaI based on the Renyi divergence, which improves our theoretical understanding on CaI.
> Furthermore, the fact that the theoretically established our framework (RCaI) gives risk-sensitive extension of the well-known SAC also strengthens our contribution.
>
> Additionally, we would like to emphasize that the derived RSAC algorithm requires only a minor modification to the standard SAC. Indeed, by letting $ \eta = 0 $ in the gradients (34)-(36), we recover SAC. This is an advantage of RSAC because techniques for stabilizing SAC, e.g., reparameterization, target networks, double Q-network, can be directly used for RSAC.
>
> We will specify the above in the revised version.
>
>
>
>
> > One way to justify the proposed method . . . MPO, VMBPO, Mismatched no More, etc.
>
> We sincerely apologize for the lack of explanation.
>   The notion of risk of the risk-sensitive control with exponential utility can be described by minimax robust control, and through the experiment, we aim to show the robustness of policies learned by RSAC for risk-averse cases $ \eta > 0 $.
>   Indeed, for the unregularized case, it is known that risk-sensitive control with $ \eta > 0 $ equivalently solves the following minimax control problem (Petersen, James, Dupuis, (IEEE TAC, 2000)): $$ \\min_{\\{u_t\\}} \max_{\\nu \\in B_d (\\mu)} \\mathbb{E}\_{\\nu} [ c_T(x_T) +  \\sum_{t=0}^{T-1} c_t(x_t,u_t) ] ,$$
>  where $ \mu $ is the reference distribution of the trajectory $ \\{x\_t\\}\_{t=0}^{T} $, $ \\nu $ is the perturbed distribution of $ \\{x\_t\\}\_{t=0}^{T} $, and $ B_d (\\mu) := \\{ \\nu : D_{\\rm KL} (\\nu \\| \\mu) \\le d \\} $ defines the set of all admissible perturbed distributions $ \\nu $. The radius $ d $ is related to the sensitivity parameter $ \\eta $. This minimax problem optimizes the control considering the worst-case perturbation of the state distributions, which means that the risk-sensitive control with $ \eta > 0 $ is robust against the perturbation of system parameters and the noise.
>   Therefore, RSAC is expected to learn robust policies. However, we have not yet revealed the equivalence between the "regularized" risk-sensitive control and minimax control. The role of the experiment in this paper is to verify the robustness of RSAC, which has not yet been ensured theoretically in this work.
>   Consequently, we have observed the robustness as expected.
>   To make the robustness clearer, we have plotted the empirical distribution of the cost for different $ \\eta $ in Fig. 1 of the attached PDF. Please see the PDF for the discussion of this additional evaluation. It should be noted that the experiment shows RSAC with $ \\eta > 0 $ is more robust than the standard SAC ($ \\eta = 0 $).
>
>   We believe that the experiment together with this additional evaluation plays a sufficient role in complementing the theoretical contribution of this work.
>   Even though it would be preferable to compare RSAC with methods other than SAC to show its advantages besides the robustness, it is beyond the scope of this work because the focus of this work is on a theoretically established unifying framework of CaI, not on RSAC. We plan to study the properties of RSAC including its numerical issue in a forthcoming paper; please see also the response to the comment on the numerical instability.
>
> We will specify the above in the revised manuscript.
>
>
> > L284-286 : It is unclear what the authors intend . . .  What are you showing?
>
> We apologize for the unclear explanation. As explained above, the experiment is conducted to verify the robustness of policies learned by RSAC.
>
>
> > The existence of the exponential function  . . . beyond noting it in the conclusion.
>
> First, we would like to emphasize that in the experiment, even for $ \\eta = 0.02 $, where the numerical instability is not problematic, the robustness of the learned policy is improved. Hence, although it is desirable to resolve the numerical instability for large $ |\\eta| $, we consider RSAC to be useful enough at the moment to improve the robustness of the policies.
>
> It may be possible to alleviate this numerical issue by considering a dual problem associated with the regularized risk-sensitive control as done for resolving the numerical instability of the Sinkhorn algorithm used for solving entropic optimal transport problems.
> We would like to try this approach in a forthcoming paper, where we aim to reveal the properties of RSAC.
> Lastly, we would like to emphasize that this issue is not specific to our algorithms, but occurs in general risk-sensitive RL with exponential utility.
>
>
> > L147-151 : These are known results . . . provided for them
>
> We would like to cite (Whittle, (John Wiley & Sons, 1990)) for them.

---

> > ### Comment · Reviewer_qM7b · 2024-08-10
> > **Thanks for the responses**
> >
> > Thank you for the thorough responses.  I reviewed my and your comments, as well as re-read the paper and in retrospect feel my score of 2 was a bit harsh.  I will upgrade my score to a 3.
> >
> > That said, I think this paper presents nice preliminary work but has some fundamental issues.  First, lack of experimental validation.  I appreciate the authors' position that this is intended to be a theoretical work, but fundamentally you propose an algorithm and then fail to demonstrate that it is effective in any reasonable settings other than one very limited experiment.  All other reviewers call out this weakness and, surprisingly in my opinion, seem to overlook it in their scoring.  **I would urge other reviewers to reconsider the suitability of this paper for NeurIPS with the present state of experimental validation or lack thereof.** The work would have much higher impact if the resulting algorithm were more extensively validated.
> >
> > The second fundamental weakness is that this paper is poorly motivated.  The authors give no motivation as to why one would consider Renyi divergence in place of KL divergence beyond citing a few papers that have looked into this.  The authors imply that derivation of the exponential risk measure is a contribution, but the equivalence of CaI (or equivalently RL-as-inference) and the exponential utility is well-known.    Here are a few papers that show this relationship:
> >
> > * Equation (8) : O'Donoghue, Brendan. "Variational bayesian reinforcement learning with regret bounds." Advances in Neural Information Processing Systems 34 (2021): 28208-28221.
> >
> > * Appendix A.1 : Eysenbach, Benjamin, et al. "Mismatched no more: Joint model-policy optimization for model-based rl." Advances in Neural Information Processing Systems 35 (2022): 23230-23243.
> >
> > * Noorani, Erfaun, Christos Mavridis, and John Baras. "Risk-sensitive reinforcement learning with exponential criteria." arXiv preprint arXiv:2212.09010 (2022).

---

> > > ### Author Response · Authors · 2024-08-12
> > > **Thank you for your additional comments**
> > >
> > > Thank you for introducing the papers. Nevertheless, they do not provide the equivalence between CaI and risk-sensitive control with exponential utility in a satisfactory manner. Detailed explanations are as follows.
> > >
> > > > Equation (8) : O'Donoghue, Brendan. "Variational bayesian reinforcement learning with regret bounds." Advances in Neural Information Processing Systems 34 (2021): 28208-28221.
> > >
> > > Equation (8) of the above paper is a well-known result that for a given function $ K_l^t $, the optimal value of the entropy-regularized optimization problem takes the form of the exponential utility. However, the fact that the optimal value is given by the exponential utility does not imply the equivalence between the MaxEnt control (CaI using the KL divergence) and the risk-sensitive control. Although the exponential utility function is utilized in this paper, the risk-sensitive control problem is not addressed, and thus, the relationship between CaI and the risk-sensitive control is not revealed at all.
> > >
> > >
> > > > Appendix A.1 : Eysenbach, Benjamin, et al. "Mismatched no more: Joint model-policy optimization for model-based rl." Advances in Neural Information Processing Systems 35 (2022): 23230-23243.
> > >
> > > As mentioned in Appendix B.1 (A.1 may be incorrect) of the above paper, VMBPO solves a risk-seeking control problem with exponential utility. For the variational inference of the distribution of the state and control input trajectory, VMBPO uses the variational distribution whose transition distribution is not fixed. This setting essentially results in the risk-**seeking** (optimistic) policies as known in (S. Levine, arXiv:1805.00909, 2018). However, we would like to emphasize that from this setting, we cannot derive the equivalence between CaI and risk-**averse** control problems. Our result does not have this restriction, which implies that for connecting the risk-sensitive control and probabilistic inference, CaI using the Renyi divergence is a more appropriate framework.
> > >
> > >
> > > > Noorani, Erfaun, Christos Mavridis, and John Baras. "Risk-sensitive reinforcement learning with exponential criteria." arXiv preprint arXiv:2212.09010 (2022).
> > >
> > > In the above paper, the equivalence between a risk-sensitive control problem and a maxmini control problem with KL regularization is shown (Corollary 1). Although this result is interesting, it is not related to the equivalence between CaI and the risk-sensitive control. Additionally, in Remark 1, it is mentioned that by making "heuristic assumptions" on the measure $P_\mu$, the MaxEnt control objective can be reconstructed by the risk-sensitive control objective. However, these heuristic assumptions suppose that the distribution of the optimal trajectory of the state and the action is uniform, which is not satisfied in general.
> > > Our equivalence result does not require such unrealistic assumptions. This fact also clarifies that for connecting the risk-sensitive control and probabilistic inference, CaI using the Renyi divergence is a more appropriate framework.
> > > We also would like to emphasize that the above paper does not show the relationship between the risk-sensitive control and probabilistic inference.
> > >
> > > We would like to add the above explanations to the revised version.
> > > If there are any other papers we should comment on, we would be grateful if you could share them with us.
> > >
> > >
> > > If you understand our equivalence results (please see Fig. 1), you will also understand their importance, and the above responses further clarify our theoretical contributions of extending CaI to RCaI. Our approach using the Renyi divergence enables the risk-sensitive extension of CaI, which cannot be attained by the previous approaches as explained above.
> > > If you still think our motivation is weak even considering the above responses, we do not know what motivation is needed beyond the following facts:
> > > - Using a divergence other than the KL divergence in model predictive control as inference gives good experimental results (Wang, So, Gibson, et. al., in Robotics: Science and Systems, 2021).
> > > - Variational inference using divergences other than the KL divergence has been well studied in the machine learning community.
> > >
> > >
> > > > The work would have much higher impact if the resulting algorithm were more extensively validated.
> > >
> > > We agree with this, and we are sorry that we were not able to conduct additional experiments. However, we would like to emphasize again that the proposal of the RL algorithms is a byproduct of RCaI. Experiments are not essential to demonstrate the correctness and the significance of the theoretical results.
> > > Even though there is room for additional evaluations of the algorithms, we believe our theoretical contributions are enough to be considered for publication in NeurIPS, which publishes many theoretical papers.

---

> ### Comment · Reviewer_qM7b · 2024-08-12
> **Thanks for the responses**
>
> Thanks for taking the time to respond to my concerns.  Due to my reviewer load I do not have time to respond to all of your comments, but will respond to the high-priority items.
>
> > "This setting essentially results in the risk-seeking (optimistic) policies...we cannot derive the equivalence between CaI and risk-averse control problems."
>
> The authors show the (well-known) equivalence to the entropic risk objective.  Equivalence to the entropic risk objective is sufficient to show, both, risk-seeking and risk-averse control.  Note that the entropic risk objective is the cumulant generating function and thus (by Taylor series) is equivalent to (in your notation and as shown on L148-149):
>
> $$\frac{1}{\eta} \log \mathbb{E}[\exp(\Phi(\tau))] = \mathbb{E}[\Phi(\tau)] + \frac{\eta}{2} Var[\Phi(\tau)] + O(\eta^2)$$
>
> From this equivalence it is clear that positive $\eta$ biases towards policies that have higher return variance (i.e. risk-seeking) and negative $\eta$ biases towards policies with lower return variance (i.e. risk-averse).  Discussion in the CaI (and RL-as-inference) literature is centered on risk-seeking policies because they are more problematic, but risk-averse policies are easily obtainable from the formulation.
>
> > "We agree with this, and we are sorry that we were not able to conduct additional experiments."
>
> The present state of experimental validation is far below the standard set by NeurIPS, despite this being a theoretically oriented paper.  This opinion is shared by all reviewers, but we differ in how we consider the impact of this in our scoring.  I appreciate the nature of tight deadlines but, at the end of the day, the experimental validation is simply below standard in my opinion and I cannot argue for acceptance in good conscience.  That said, I do think this line of work has merit and should be further developed.

---

> > ### Author Response · Authors · 2024-08-13
> > **Thank you again for your response**
> >
> > Thank you for your response despite your reviewer load.
> >
> > >  Equivalence to the entropic risk objective is sufficient to show, both, risk-seeking and risk-averse control.
> >
> > We would like to explain that this comment is incorrect and that the difference between the previous work and ours is significant, which means RCaI is the appropriate risk-sensitive extension of CaI.
> > VMBPO maximizes the entropic risk objective $ \frac{1}{\eta} \log \mathbb{E} [\exp(\eta \Phi(\tau))] $. However, we would like to emphasize that the equivalence for VMBPO holds **only for positive $\eta$ resulting in risk-seeking policies**. Tricks such as the change of variables do not work to obtain risk-averse policies by VMBPO. Risk-averse policies ($\eta < 0$) cannot be obtained by their approach inherently. The risk-seeking property of the policies obtained by CaI whose variational distribution does not fix its transition distribution, is intrinsic.
> > This is why we mentioned that we cannot derive the equivalence between CaI and risk-averse control problems from VMBPO.
> > To derive the equivalence between CaI and risk-sensitive control for both risk-seeking and risk-averse cases, we need a fundamentally different approach from the previous work, and the solution we discovered is CaI using the Renyi divergence.
> > Our equivalence can deal with both risk-seeking ($\eta > 0$) and risk-averse ($\eta < 0$) policies. This is a significant difference between the previous work and our result. This difference is crucial because the robustness of risk-averse policies is important for applications.
> >
> > For reference, we would like to note that in Appendix B.1 of the following suggested paper, the risk-sensitivity parameter $ \eta $ is restricted to be positive (risk-seeking). This is not for simplicity, but negative sensitivity (risk-averse) parameters cannot be dealt with in the framework used by VMBPO.
> > > Eysenbach, Benjamin, et al. "Mismatched no more: Joint model-policy optimization for model-based rl." Advances in Neural Information Processing Systems 35 (2022): 23230-23243.

---

### Official Review · Reviewer_pMhb · 2024-07-12

**Soundness:** 3
**Presentation:** 4
**Contribution:** 3
**Rating:** 6
**Confidence:** 4

**Summary:**

The paper generalizes the control as inference framework to the risk-sensitive setting using Renyi divergence variational inference. This yields a cost function with an exponential utility and log-probability regularization, weighted by the Renyi divergence parameter $\eta$. From the Taylor expansion of the cost, we can see that $\eta$ controls the level to which we are risk-averse or risk-seeking. And when $\eta$ goes to zero, we recover the traditional MaxEnt control problem. Next, the authors show that the risk-sensitive optimal policy can be obtained by solving the soft Bellman equation, which relates it to many existing methods. They also show that for deterministic dynamics, their framework, RCaI, and MaxEnt give the same optimal policy. The authors then develop RCaI versions of the policy gradient and soft actor-critic algorithms. They also show that using Renyi entropy regularization in place of the standard KL divergence in maximum entropy control yields an optimal policy with the same structure. Finally, they provide a proof-of-concept experiment on the classical Pendulum benchmark using their risk-sensitive SAC algorithm. They show that the choice of risk-sensitivity parameter can improve robustness to dynamics mismatch.

**Strengths:**

- RCaI is a novel formulation of risk-sensitive control using the control-as-inference framework with a unique choice of divergence metric. They derive two RL algorithms from this framework, a policy gradient and soft actor-critic algorithm, which are easily swapped in for their risk-neutral equivalents.
- The authors show that RCaI actually generalizes the MaxEnt formulation to incorporate risk sensitivity. They also show many interesting properties of the optimal policy and how it relates to existing approaches.
- They show experimentally that RCaI may yield improvements in robustness by introducing the risk-sensitivity term.
- The paper is well organized and overall written well. It provides a thorough related work section and does a good job explaining the novelty and results.

**Weaknesses:**

- The experimental evaluation is sparse, with only one simple task and no baselines other than RSAC with $\eta=0$. They also do not evaluate the policy gradient method and contrast it with RSAC.

**Questions:**

- How does the risk-sensitive policy gradient method compare to RSAC and REINFORCE?

**Limitations:**

The authors discuss limitations of their method in the conclusions. Specifically, they discuss the numerical instability of their method for large $\eta$ cases.

---

> ### Author Rebuttal · Authors · 2024-08-07
>
> Thank you very much for your thoughtful comments.
>
> > The experimental evaluation is sparse, with only one simple task and no baselines other than RSAC with $ \eta = 0 $. They also do not evaluate the policy gradient method and contrast it with RSAC.
>
> We sincerely apologize for the lack of explanation.
> First, we would like to clarify the purpose of the experiment in this work.
>   The notion of risk of the risk-sensitive control with exponential utility can be described by minimax robust control, and through the experiment, we aim to show the robustness of policies learned by RSAC for risk-averse cases $ \\eta > 0 $.
>   Indeed, for the unregularized case, it is known that risk-sensitive control with $ \\eta > 0 $ equivalently solves the following minimax control problem [R1]: $$ \\min_{\\{u_t\\}} \\max_{\nu \\in B_d (\\mu)} \\mathbb{E}\_{\\nu} [ c_T(x_T) +  \\sum_{t=0}^{T-1} c_t(x_t,u_t) ], $$
> where $ \\mu $ is the reference distribution of the trajectory $ \\{x_t\\}\_{t=0}^{T} $, $ \nu $ is the perturbed distribution of $ \\{x_t\\}\_{t=0}^{T} $, and $ B_d (\\mu) := \\{ \\nu : D_{\\rm KL} (\\nu \\| \\mu) \\le d \\} $ defines the set of all admissible perturbed distributions $ \\nu $. The radius $ d $ is related to the sensitivity parameter $ \\eta $. This minimax problem optimizes the control considering the worst-case perturbation of the state distributions, which means that the risk-sensitive control with $ \\eta > 0 $ is robust against the perturbation of system parameters and the noise.
>   Therefore, RSAC is expected to learn robust policies. However, we have not yet revealed the equivalence between the "regularized" risk-sensitive control and minimax control. The role of the experiment in this paper is to verify the robustness of RSAC, which has not yet been ensured theoretically in this work.
>   Consequently, we have observed the robustness as expected.
>
> As an additional evaluation, we have plotted the empirical distributions of the cost under RSAC for different $ \\eta $ in Fig. 1 of the attached PDF. As can be seen, only the distribution for $ \\eta = 0.02 $ does not change so much under the system perturbations. The distribution for SAC ($ \\eta = 0 $) with $ l = 1.5 $ deviates from the original one ($ l = 1.0 $), and another peak of the distribution appears in the high-cost area. This means that there is a high probability of incurring a high cost, which clarifies the advantage of RSAC.
> On the other hand, the more risk-seeking the policy becomes, the less robust against the system perturbation it becomes. However, at the expense of the robustness, for $ l = 1.25 $, the policy with $ \\eta = -0.02 $ yields a high probability in the low-cost area (from cost $ =20 $ to $ {\rm cost} = 80 $).
>
>
> We believe that the experiment together with this additional evaluation plays a sufficient role in complementing the theoretical contribution of this work.
> Even though it would be preferable to compare RSAC with any other methods to show its advantages besides the robustness, it is beyond the scope of this work because our main motivation is to extend the framework of CaI and to provide new theoretical perspectives on CaI.
> We plan to study the properties of RSAC in a forthcoming paper.
>
>
> [R1] I. R. Petersen, M. R. James, and P. Dupuis, "Minimax optimal control of stochastic uncertain systems with relative entropy constraints," IEEE Transactions on Automatic Control, vol. 45, no. 3, 2000.
>
>
>
> > How does the risk-sensitive policy gradient method compare to RSAC and REINFORCE?
>
> We are very sorry that we were not able to conduct the additional experiment of the derived risk-sensitive REINFORCE in time for the rebuttal.
> However, we would like to mention that REINFORCE suffers from high variance, delayed updates, and less sample efficiency, and SAC generally outperforms REINFORCE in a wide range of environments. Hence, it is expected that RSAC also outperforms the derived risk-sensitive REINFORCE. For the comparison between the risk-sensitive REINFORCE and the standard REINFORCE, the previous work [20] showed that by using an appropriate sensitivity parameter $ \\eta $, the risk-sensitive REINFORCE learns faster and has lower variance than the standard REINFORCE for the unregularized case.
> The regularized risk-sensitive REINFORCE derived in this paper will have similar properties to the unregularized risk-sensitive REINFORCE because the only difference between the regularized and unregularized REINFORCE methods is the presence of the log-probability term in the policy gradient.
>
> [20] E. Noorani and J. S. Baras, "Risk-sensitive REINFORCE: A Monte Carlo policy gradient algorithm for exponential performance criteria," in 2021 60th IEEE Conference on Decision and Control (CDC), pp. 1522-1527, 2021.

---

### Official Review · Reviewer_4yoD · 2024-07-14

**Soundness:** 4
**Presentation:** 3
**Contribution:** 3
**Rating:** 6
**Confidence:** 4

**Summary:**

This paper considers the control as inference framework of RL. Instead of minimizing the KL which is commonly done in control as inference (and has been shown to be equivalent to MaxEnt RL), they consider minimizing the Rényi divergence. They prove that this minimization is equivalent to minimizing a functional similar to the entropic risk measure, and show that the order of the Rényi divergence controls the risk behaviour of the policy obtained (averse/neutral/seeking). They then show that this minimization over policies is equivalent to solving a soft Bellman equation. They then take another route of generalizing maximum entropy control, by replacing the standard entropy with Rényi entropy, which they prove is also optimized by the same soft Bellman equation. They then show how this can be transformed into an implementable algorithm, by introducing a variant of policy gradient and soft actor-critic for their formulation. They round out their work with an experiment section to illustrate properties of their algorithm at varying levels of risk sensitivity.

**Strengths:**

I find this paper to be clearly written, and guides the reader through their contributions. The theoretical results and their proof also flow nicely and are clearly presented.

I think that taking the two paths to generalize MaxEnt RL (replacing the minimization of the KL with Rényi and replacing the entropy regularization with Rényi entropy regularization) is a very nice approach and makes a strong case for why this is a natural generalization.

Overall I think this is an interesting theoretical work and extension to the MaxEnt RL framework, and I think it can inspire future work.

**Weaknesses:**

I believe that the major weakness of this work is the experiment section. I understand the goal of this work is mainly theoretical, however, I believe the current experiments are unmotivated, and do not complement the theoretical results. In particular, there is currently only a single experiment, which I believe to be unfit for the following reasons:
- the experiment measures the generalization across environments of the algorithm at different levels of $\eta$. I am quite surprised by this,  as none of the theoretical results discuss generalization across environments.
- the theoretical results prove that the algorithm optimizes a quantity that *does not* solely depend on the expectation of the cost function, but the plot shows the average cost obtained by the algorithm.
- touching on the previous point, I think that what the experiment section can be quite helpful for, which it currently lacks, is provide the reader with a more intuitive understanding of the level of risk sensitivity as a function of $\eta$. For example, perhaps one could plot the empirical distributions of returns obtained for different $\eta$ (so that one can visualize the difference between risk-seeking/neutral/averse, and how continuous this behaviour is wrt $\eta$).
- there is only a single environment used (Pendulum-v1), with no justification as to why this was chosen.

I believe that if the experiment section can be improved in the ways I highlighted above, the paper has the potential to be more interpretable and impactful, and I would be happy to update my score to reflect this.

**Questions:**

- It is written that for large $|\eta|$ the algorithm is unstable. In the experiments, the largest $|\eta|$ used is .02. Is the algorithm unstable for $\eta$ larger than this? A better understanding of what range $\eta$ can safely take would be helpful.
- There have been a number of risk-senstive soft actor critic algorithms proposed in recent years, none of which are referenced in this paper. Is there a reason why you haven't mentioned them, and compared/contrasted your proposed algorithm to them?

**Limitations:**

The authors adequately discuss the limitations, and list them as items for future work (such as numerical instability for large $|\eta|$).

---

> ### Author Rebuttal · Authors · 2024-08-07
>
> Thank you very much for your thoughtful comments.
>
>
> > the experiment measures the generalization . . . discuss generalization across environments.
>
> We apologize for the lack of explanation.
>   Through the experiment, we aim to show the robustness of policies learned by the risk-sensitive soft actor-critic (RSAC) for risk-averse cases $ \\eta > 0 $.
>   Indeed, for the unregularized case, it is known that risk-sensitive control with exponential utility solves a robust control problem (Petersen, James, Dupuis, (IEEE TAC, 2000)).
>   Specifically, risk-sensitive control with $ \\eta > 0 $ equivalently solves the following minimax control problem: $$ \\min_{\\{u_t\\}} \\max_{\\nu \\in B_d (\\mu)} \mathbb{E}\_{\\nu} [ c_T(x_T) +  \\sum_{t=0}^{T-1} c_t(x_t,u_t) ], $$
> where $ \\mu $ is the reference distribution of the trajectory $ \\{x_t\\}\_{t=0}^{T} $, $ \\nu $ is the perturbed distribution of $ \\{x_t\\}\_{t=0}^{T} $, and
> $ B_d (\mu) := \\{ \\nu : D_{\\rm KL} (\\nu \\| \\mu) \\le d \\} $ defines the set of all admissible perturbed distributions $ \\nu $. The radius $ d $ is related to the sensitivity parameter $ \\eta $. This minimax problem optimizes the control considering the worst-case perturbation of the state distributions, which means that the risk-sensitive control with $ \\eta > 0 $ is robust against the perturbation of system parameters and the noise.
>   Therefore, RSAC is expected to learn robust policies, which generalize well across environments. However, we have not yet revealed the equivalence between the "regularized" risk-sensitive control and minimax control.
>   The role of the experiment in this paper is to verify the robustness of RSAC, which has not yet been ensured theoretically in this work.
>   Consequently, we have observed the robustness as expected.
>
>   We will add the above explanation to the revised manuscript.
>
>
> > the theoretical results prove that  . . . one could plot the empirical distributions of returns obtained for different $ \eta $
>
> Following your suggestion, we have plotted the empirical distributions of costs for different $ \eta $ in Fig. 1 of the attached PDF. As can be seen, only the distribution for $ \eta = 0.02 $ does not change so much under the system perturbations. The distribution for SAC ($ \eta = 0 $) with $ l = 1.5 $ deviates from the original one ($ l = 1.0 $), and another peak of the distribution appears in the high-cost area. This means that there is a high probability of incurring a high cost, which clarifies the advantage of RSAC.
> On the other hand, the more risk-seeking the policy becomes, the less robust against the system perturbation it becomes. However, at the expense of the robustness, for $ l = 1.25 $, the policy with $ \eta = -0.02 $ yields a high probability in the low-cost area (from cost $ =20 $ to $ {\rm cost} = 80 $).
> We will add the evaluation of the empirical distributions to the revised manuscript. We appreciate your suggestion.
>
>
>
>
> > there is only a single environment used (Pendulum-v1), with no justification as to why this was chosen.
>
> We are very sorry that we were not able to conduct an additional experiment on a different environment in time for the rebuttal.
>   As mentioned above, the purpose of the experiment is to verify the robustness of RSAC, which has not been proved theoretically in this paper. Although it would be preferable to use several environments, we believe that the experiment together with the additional evaluation of the empirical distributions plays a sufficient role in complementing the theoretical contribution of this work.
>
> > I believe that if the experiment section can be improved in the ways I highlighted above, the paper has the potential to be more interpretable and impactful, and I would be happy to update my score to reflect this.
>
> We hope the additional evaluation and the responses have made our work more interpretable and impactful.
>
>  > It is written that for large $ |\eta| $
> the algorithm is unstable. . . . A better understanding of what range $ \eta $
> can safely take would be helpful.
>
> Since $ \eta $ appears, for example, as $ \exp(\eta Q^{(\phi)}(x_t,u_t)) $ in the gradients (34)-(36), the magnitude of $ \eta $ that does not cause the numerical instability depends on the scale of the reward (cost).
>   Therefore, we need to choose $ \eta $ depending on environments.
>   In the experiment using Pendulum-v1, $ |\eta| $ that is larger than $ 0.03 $ results in the failure of learning due to the numerical instability. We would like to emphasize that this issue is not specific to our algorithms, but occurs in general risk-sensitive reinforcement learning with exponential utility.
>
>   Thank you for your suggestion. We will add the above explanation to the revised version.
>
>
> > There have been a number of risk-sensitive soft actor critic algorithms . . . Is there a reason why you haven't mentioned them, and compared/contrasted your proposed algorithm to them?
>
>
>
> This is because in this paper, we focus on the risk sensitivity induced by the exponential performance criteria.
>   To the best of our knowledge, the only work that proposes a risk-sensitive soft actor-critic type algorithm for the exponential utility is (Enders, Harrison, Schiffer, (arXiv:2402.09992, 2024)) mentioned in the manuscript.
>   If you know of any other references and could share them with us, we sincerely appreciate it.
>
>   If we should mention another type of risk-sensitive soft actor-critic, we would like to add the references (Duan, Guan, Li, Ren, Sun, Cheng, (IEEE TNNLS, 2021), (Choi, Dance, Kim, Hwang, Park, (ICRA, 2021)), which consider risk by distributional RL, to the manuscript.
>   An advantage of RSAC over other risk-sensitive approaches including the distributional RL is that we only need minor modifications to the standard SAC. Thanks to this, techniques for stabilizing SAC, e.g., reparameterization, minibatch sampling with a replay buffer, target networks, double Q-network, can be directly used for RSAC.

---

### Official Review · Reviewer_5JeB · 2024-07-16

**Soundness:** 3
**Presentation:** 3
**Contribution:** 3
**Rating:** 7
**Confidence:** 3

**Summary:**

In this paper, the authors consider a risk-sensitive control problem with Renyi divergence. The contributions are primarily theoretical with some rudimentary experiments. The contributions include connection to risk-sensitive control under exponential cost formulation, with Renyi divergence leading to an additional regularization term. Extensions to a RL setting with a policy gradient theorem and an actor-critic version are proposed. An alternative risk-senstive control problem with Renyi
entropy is discussed.

**Strengths:**

- Tackles an important risk-sensitive control problem and makes advances by employing a general divergence measure
- Connections to exponential utility formulation

**Weaknesses:**

- Some bits of RL extension are unclear to me (see questions below).

**Questions:**

1. The policy gradient theorem in Prop 7 has an expectation over trajectories. Does this mean for one update to the policy parameter, an entire trajectory needs to be simulated? In other words, is there a REINFORCE variant that updates after every sample transition?
2. The soft-actor critic extension is unclear to me. Does the proposed soft actor critic algorithm with gradient estimates in (34)-(36)? If yes, can you characterize the limiting policy wrt risk-sensitivity? What about compatibility issues that one usually sees with actor-critic algorithms that employ function approximation?
3. While this may be outside the scope of this work, can you comment on extensions to a long-run average cost formulation, with Renyi divergence?

---

> ### Author Rebuttal · Authors · 2024-08-07
>
> Thank you very much for your thoughtful comments. We sincerely appreciate your positive evaluation.
>
> > The policy gradient theorem in Prop 7 has an expectation over trajectories. Does this mean for one update to the policy parameter, an entire trajectory needs to be simulated? In other words, is there a REINFORCE variant that updates after every sample transition?
>
> We are not entirely certain whether the derived risk-sensitive REINFORCE can be extended to the one that updates the policy parameter after every sample transition because we do not know of the specific literature that the reviewer has in mind.
>   Nevertheless, we expect that such extension is possible because the policy gradient (23) for the regularized risk-sensitive control is structurally the same as the standard policy gradient for the risk-neutral control.
>   That is, by replacing the exponential of the accumulated cost $ \\exp(\\eta c_T(x_T) + \\eta \\sum_{s=t}^{T-1} (c_s(x_s,u_s) + \\log \\pi^{(\\theta)} (u_s | x_s))) $ in (23) by $  c_T(x_T) +  \\sum_{s=t}^{T-1} (c_s(x_s,u_s) + \\log \\pi^{(\\theta)} (u_s | x_s)) $, we recover the standard REINFORCE with regularization.
>
>
> > The soft-actor critic extension is unclear to me. Does the proposed soft actor critic algorithm with gradient estimates in (34)-(36)? If yes, can you characterize the limiting policy wrt risk-sensitivity?
>
> Yes, the proposed soft actor-critic estimates the gradient by the sample approximation of (34)-(36). For example, the gradient estimate of $ \\nabla_\\phi \mathcal{J}\_Q(\\phi) $ is given by $ (\\nabla_\\phi Q^{(\\phi)}(x_t,u_t)) \\exp(\\eta Q^{(\\phi)}(x_t,u_t) - \\eta c(x_t,u_t))  \\{ T_\\eta (Q^{(\\phi)}(x_t,u_t) - c(x_t,u_t))  -  T_\\eta( V^{(\\psi)}(x_{t+1}) )  \\}  $, where $ x_t $, $ u_t $, and $ x_{t+1} $ are samples.
>
>
>   Although we do not yet know how to analyze the limiting policy learned by the proposed soft actor-critic algorithm, it is interesting to investigate the relationship between the risk-sensitivity parameter $ \eta $ and the robustness of the limiting policy. We appreciate your insightful comment, and we would like to study it in future work.
>
>
>
> > What about compatibility issues that one usually sees with actor-critic algorithms that employ function approximation?
>
> The compatible function approximation theorem can be extended to the proposed soft actor-critic algorithm.
>   For the standard actor-critic algorithm, the compatibility of a function approximator $ Q^{(\\phi)} $ is defined by the condition $ \\nabla_\\phi Q^{(\\phi)} (x,u) =  \\nabla_\\theta \\log \\pi^{(\\theta)} (u_t|x_t) $ [R1]. For the proposed risk-sensitive soft actor-critic (RSAC), the compatibility condition is modified as $ \\nabla_\\phi Q^{(\\phi)} (x,u) \\exp(\\eta Q^{(\\phi)} (x,u) - \\eta c(x,u)) =  \\nabla_\\theta \\log \\pi^{(\\theta)} (u_t|x_t) $, where $ \\eta $ is the risk-sensitivity parameter. By letting $ \\eta = 0 $, we recover the standard compatibility condition.
>   We would like to omit the detail of the compatible function approximation theorem for RSAC because we plan to study the properties of RSAC in a forthcoming paper.
>
> [R1] R. S. Sutton, D. McAllester, S. Singh, and Y. Mansour, "Policy gradient methods for reinforcement learning with function approximation," Advances in Neural Information Processing Systems, vol. 12, pp. 1057-1063, 1999.
>
>
>
>
>  > While this may be outside the scope of this work, can you comment on extensions to a long-run average cost formulation, with Renyi divergence?
>
> There will be several technical difficulties in dealing with the regularized risk-sensitive control (reinforcement learning) for the long-run average cost by the same reason as for the unregularized case [29, R2]. Even so, we expect that similar results to ours will hold for the long-run average problems, that is, the regularized risk-sensitive control problem with long-run average cost will boil down to solving a soft Bellman equation, and this leads to risk-sensitive soft actor-critic for long-run average cost.
> We would like to tackle this important and challenging extension in future work.
>
>
> [29] V. S. Borkar, "Q-learning for risk-sensitive control," Mathematics of Operations Research, vol. 27, no. 2, pp. 294-311, 2002.
>
> [R2] A. Biswas and V. S. Borkar, "Ergodic risk-sensitive control--a survey," Annual Reviews in Control, vol. 55, pp. 118-141, 2023.

---

### Author Rebuttal · Authors · 2024-08-07

We have attached a PDF file with the experimental results of the empirical distributions of cost under the derived risk-sensitive soft actor-critic.

---

### Decision · Program_Chairs · 2024-09-25

**Decision:**

Accept (poster)

**Comment:**

The paper investigates risk-sensitive control using Renyi divergence in the control-as-inference (CaI) framework. The authors establish a connection between Renyi divergence and risk-sensitive control, with theoretical results showing that the Renyi order parameter controls risk behavior (averse/neutral/seeking). They propose a risk-sensitive policy gradient method and a soft actor-critic (RSAC) variant based on this framework. While most of the reviewers agree with the novelty of this approach, some of the reviewers raise concerns about the experiments. The current experiment results seem a bit toyish. It would be great if the authors could include more experiments in the next version.